biomimetics/robotics/biomechanics

robotic insects, design, flapping wings, DC motors, resonance, scalability

**Author for correspondence:**
Mostafa R. A. Nabawy
e-mail: mostafa.ahmednabawy@manchester.ac.uk

# Scalability of resonant motor-driven flapping wing propulsion systems

Mostafa R. A. Nabawy[1,2] and Ruta Marcinkeviciute[1]

[1]Department of Mechanical, Aerospace and Civil Engineering, The University of Manchester, Manchester M1 3BB, UK
[2]Aerospace Engineering Department, Faculty of Engineering, Cairo University, Giza 12613, Egypt

MRAN, 0000-0002-4252-1635

This work aims to develop an integrated conceptual design process to assess the scalability and performance of propulsion systems of resonant motor-driven flapping wing vehicles. The developed process allows designers to explore the interaction between electrical, mechanical and aerodynamic domains in a single transparent design environment. Wings are modelled based on a quasi-steady treatment that evaluates aerodynamics from geometry and kinematic information. System mechanics is modelled as a damped second-order dynamic system operating at resonance with nonlinear aerodynamic damping. Motors are modelled using standard equations that relate operational parameters and AC voltage input. Design scaling laws are developed using available data based on current levels of technology. The design method provides insights into the effects of changing core design variables such as the actuator size, actuator mass fraction and pitching kinematics on the overall design solution. It is shown that system efficiency achieves peak values of 30–36% at motor masses of 0.5–1 g when a constant angle of attack kinematics is employed. While sinusoidal angle of attack kinematics demands more aerodynamic and electric powers compared with the constant angle of attack case, sinusoidal angle of attack kinematics can lead to a maximum difference of around 15% in peak system efficiency.

## 1. Introduction

The development of small-scale insect-like flapping wing vehicles offers an exciting engineering opportunity for innovation, with applications ranging from artificial pollination to massively distributed sensing. Different approaches have been taken to replicate insect flight and increasingly sophisticated designs are being created using different actuation options. One of the most successful actuation options, for this class of vehicles, is piezoelectric actuators. In fact, piezoelectric-actuated concepts have demonstrated tethered lift-off [1–5], have achieved tethered

controlled hover [6,7], have been used to demonstrate scalable prototypes [8] and most recently have flown using onboard solar cells [9]. Designs based on other actuation options also managed to demonstrate successful tethered lift-off including electromagnetic actuators [10,11] and soft-muscle actuators [12]. On the other hand, motors have traditionally been employed to actuate relatively large and heavy flapping vehicles typically employing slider-crank transmission mechanisms (or similar) to realize the required flapping motion. However, recently, a new class of motor-actuated insect-like aerial vehicles was conceived demonstrating successful lift-off [13–16]. These designs benefited from recent technological advancements in developing relatively small and light micro-DC motors that can be operated in an alternating fashion to directly drive the flapping wings. An elastic element is used in between the motor and the wing to ensure system resonant operation. These designs produce the necessary amplified wing motion amplitudes in a relatively simple fashion rather than using complicated transmission mechanisms. In this paper, we will focus on this class of motor-driven designs that directly drive the flapping wings at the system resonance frequency.

Campolo *et al.* [17,18] considered resonant motor-driven vehicles where to predict the generated lift force and the aerodynamic power consumed, wing aerodynamics were represented using quasi-steady treatments based on blade element theory. This type of analysis provided means to evaluate the nonlinear aerodynamic damping experienced by the propulsion system without the need to consider the complexities of flapping wing transient aerodynamics. The wing motion was described based on sinusoidal representation of the flapping angle together with the oversimplified depiction of a constant angle of attack throughout the flapping cycle. The propulsion system was then modelled as a second-order mass–spring–damper system that is driven by an external torque. Of particular importance is the functionality of the spring to recover energy and to cause resonance at a particular moment of inertia of a system. To validate their theoretical model, a tethered concept that integrates micro-DC motors, elastic helical springs and artificial insect-like wings was built and tested [13]. The wing offset from the centre of flapping rotation as well as the spring stiffness were chosen as design variables for comparison between theoretical models and experimental measurements. The developed models proved useful for system performance prediction and prototype optimization, leading to a final 2.7 g prototype that successfully achieved a lift-to-weight ratio of 1.4 in an open-loop tethered lift-off test [13].

Zhang *et al.* [15], later, presented a similar modelling approach for resonant motor-driven vehicles. They provided both linear and nonlinear approaches to solve the system dynamics differential equation. By contrast to the work of Campolo *et al.* [17,18], the system moment of inertia was expanded to include gear moment of inertia; however, its significance on the dynamics model outputs may be negligible. Moreover, closed-form formulae for flapping wing energetics including total supplied power, power drained by aerodynamic damping and mean system driving efficiency were presented. However, again, their models only considered the case of the oversimplified constant angle of attack when describing the wing pitching kinematics. The theoretical predictions from both linear and nonlinear analyses were validated using both simulations and experiments on a motor-driven flapping wing vehicle. The main outputs of these comparisons proved that maximum values of lift and efficiency are achieved when driving the system at its natural frequency.

These previous studies show that integrated design of resonant flapping wing vehicles presents a particular challenge to propulsion system drivetrain design due to the need to match the dynamics of the actuator with the wing via a suitable mechanical transmission mechanism. Furthermore, for efficiency reasons, flapping wing systems must typically operate at resonance, which adds an additional layer of complexity. Without the ability to adequately model the actuator dynamics and wing aerodynamics, vehicle design has to involve a high degree of trial and error, which slows development and, therefore, only a very small part of the possible solution space can be explored.

The motivation behind this study is triggered by the need for design tools at a conceptual level to allow designers to make informed choices when developing resonant motor-driven flapping wing propulsion systems. Furthermore, the relevant design cost functions are complex, and it is necessary for a designer to be able to understand the multiple design trade-offs in order to make appropriately balanced design choices. While the models discussed above are useful for understanding potential system behaviour for given design parameters, there is a gap in understanding how this class of vehicles scale, and hence how design choices vary as the vehicle size requirements are changed. As such, this study aims to investigate the scalability of these designs as its primary objective. Moreover, previous models employed idealized wing kinematics particularly in defining the wing pitching behaviour through a constant angle of attack representation within the flapping cycle. While this is justified for the simplicity it brings to the modelling processes, it may be considered optimistic from

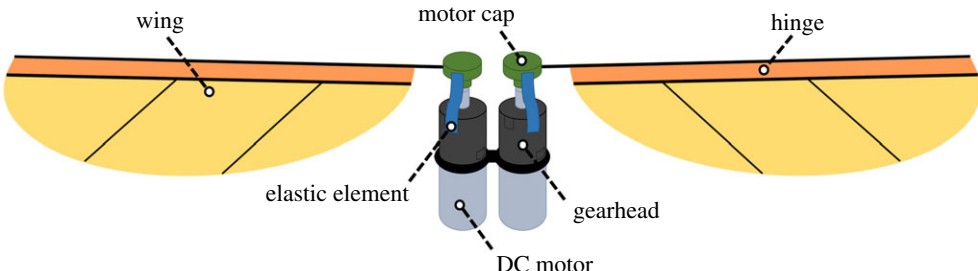

**Figure 1.** Schematic of the different components typically constituting the propulsion system of a resonant motor-direct-driven robotic insect.

an operation point of view. As such, to consider this limitation, we present additional modelling tools that consider sinusoidal variations of angle of attack allowing for a more realistic representation of the wing pitching kinematics. The outcome of this study is, therefore, a *scalable* conceptual design process that captures the interaction between electrical, mechanical and aerodynamic domains and allows designers to explore the interaction between these domains in a single transparent design environment.

The rest of the paper is arranged as follows: §2 provides details of configurational scaling considerations, aerodynamic modelling, transmission mechanism matching, actuator scaling laws and system dynamics representation, and the section ends with a simple modelling tool for comparison against rotary wing systems. Section 3 presents the results of this holistic approach for both constant and sinusoidal angle of attack kinematics. This allows to define the feasible operating conditions at different scales including frequency, elastic element stiffness and supplied voltage. Efficiency considerations are then analysed to provide a design envelope for choosing optimum system configurations. Finally, §4 discusses the findings and provides design guidelines as well as final concluding remarks.

## 2. Methods

### 2.1. Vehicle configuration

An insect-like flapping wing robot will generally contain two main systems: a propulsion system and a power system. The propulsion system typically includes wings, actuators and transmission mechanisms for matching the actuators to the wings. The required pitching motion of the wings is usually realized through a passive hinge integrated within the wing structure to ensure a less complex mechanical design. Hinge stoppers are typically included to ensure that the wing pitch angle does not exceed the optimum value for lift force generation. As for the power system, it involves an energy source in the form of a battery but could also take other forms such as a super-capacitor. The power system also includes a power electronics circuit needed to regulate the power flow from the energy source to the actuators. Hence, the power electronics circuit has two roles: (i) it converts the DC output from the power source to an AC input to the actuators ensuring that the required AC waveform variation and driving frequency are produced; and (ii) it modulates the amplitude of the AC voltage to ensure successful operation.

Figure 1 shows a schematic of an example propulsion system for a resonant motor-driven insect-like robot. The main components are the wings, transmission mechanisms (gearheads) and actuators (DC motors). The AC signal supplied to the motor ensures it flips its rotational direction generating the oscillatory flapping motion, and the gearbox transmission system immediately transfers this change in direction while modulating speed. The propulsion system also includes elastic elements for energy recovery, hence is an imperative part of this class of energetically demanding flapping vehicles. A video is provided in the electronic supplementary material for the propulsion system shown in figure 1 built using commercial off-the-shelf components. This propulsion system weighs 3.4 g. Each motor, weighing 1 g, had its gearbox, weighing 0.2 g, fully integrated to it. An elastic element in the form of an elastic band was used. The wings had an elliptic planform shape and were realized using a polyimide film spanning on top of a rigid structure of carbon fibre rods. The passive hinge required to ensure adequate wing pitching was realized using a simple way of adding an extra carbon fibre rod parallel to the leading-edge rod. As such, the membrane material enclosed between the two rods provided the hinge functionality. This method is easy in prototyping; however, the extra carbon fibre

rod adds weight to the wing structure, which is less favourable for flapping wings. Wing stoppers were added to avoid angles of attack more than 45°. This propulsion system was able to demonstrate tethered operation; however, it is important to stress that this propulsion system is included here just to demonstrate the functionality of such types of concepts. In fact, this paper will focus on developing a generic conceptual design tool for resonant motor-driven flapping propulsion systems and is not specific for a certain configuration. Currently, the propulsion system is the main technology barrier and also the one least likely to be solved by parallel developments in other microtechnology areas, hence is the focus for this work. Beyond this, attention of future studies will be focused on the other systems required for a commercially applicable product including power system, communications, sensors and control.

For the purposes of this study, it is essential to have basic mass breakdown definitions. Here, the propulsion system is considered based on the typical provision of a separate actuator for each wing. As such, we define the actuator mass fraction, $\mu_a = m_a/m_P$, representing the ratio of the mass for a *single* actuator, $m_a$, to the total mass of the propulsion systems, $m_P$. This implies that, theoretically speaking, this ratio can range from a hypothetical minimum value of zero up to a hypothetical maximum value of 0.5. However, a realistic range for this design variable is $0.25 \leq \mu_a \leq 0.45$. Note that $m_a$ and $\mu_a$ will be adopted in this study as the main design variables to assess the system performance.

## 2.2. Wing considerations

### 2.2.1. Wing planform and inertia

The two main wing planform geometric characteristics required within our models are the wing aspect ratio, AR, and wing length, $R_w$. Aspect ratio is important for aerodynamic considerations, whereas wing length is essential for defining inertia and scalability characteristics. Higher values of the wing aspect ratio are known to improve the aerodynamic induced effects [19,20]. However, as the aspect ratio increases there can be insufficient chord length to allow smooth flow reattachment on the wing upper surface. This is crucial in preventing the shedding of the leading-edge vortex and, subsequently, preventing stall on a flapping wing that would otherwise result in the wing losing its ability to generate sufficient lift coefficients. In fact, Kruyt *et al.* [21] showed that to avoid stall on revolving/flapping wings, the wing tip location measured from centre of revolution/flapping should not exceed four times the local chord. This wing tip location includes the wing length distance, $R_w$, and any existing offset distance of the wing root from the centre of revolution/flapping. It is also worth mentioning that from a structural design perspective, higher aspect ratios can lead to higher moments of inertia which directly influences the system resonance frequency. Arguably, these are the main reasons why insect wing aspect ratios are most clustered between three and four [21,22]. Thus, a constant value of 3.5 will be used for the aspect ratio in this study. Note that this aspect ratio value was recommended by Ellington for engineered vehicles [23] and was the wing aspect ratio value of the Harvard microrobotic fly [24].

Aircraft design processes require scaling laws to aid the decision-making process during the conceptual design stage [25], and the case of robotic insects is not dissimilar. For our modelling purposes, it is essential to obtain a scaling law for the wing length against the vehicle propulsion system mass. Figure 2a shows a log–log plot of the data collected for up-to-date successful designs—the data points shown in figure 2a are provided in table 1 in which the source of each data point is provided. For consistency, we have limited the employed data in table 1 and figure 2a to (i) designs that were capable of demonstrating a lift-to-weight ratio above unity; (ii) designs employing two wings, i.e. designs employing four wings are excluded; (iii) designs that rely on resonant operation to ensure relevance to the class of vehicles targeted in this study. As such, this dataset represents designs not only relying on DC motors for actuation but also other forms of actuation including piezoelectric, electromagnetic and soft muscle actuation. This is deemed appropriate within the context of finding a scaling law for top-level conceptual design purposes. It is important to note that all available design concepts are tether powered and thus the weights shown in table 1 and figure 2a only represent the propulsion system weight.

Figure 2a also illustrates the developed wing length scaling law:

$$R_w(\text{mm}) = 2.6952 m_P^{0.3727}(\text{mg}), \quad n = 12, \quad R^2 = 0.9, \quad s_{R_w/m_p} = 7.9 \text{ mm}, \tag{2.1}$$

where $n$ is the number of data points used to cast the relation and $R^2$ is the $R$-squared regression metric added to provide an estimate for the quality of the fit of the regression scaling relation. This is also

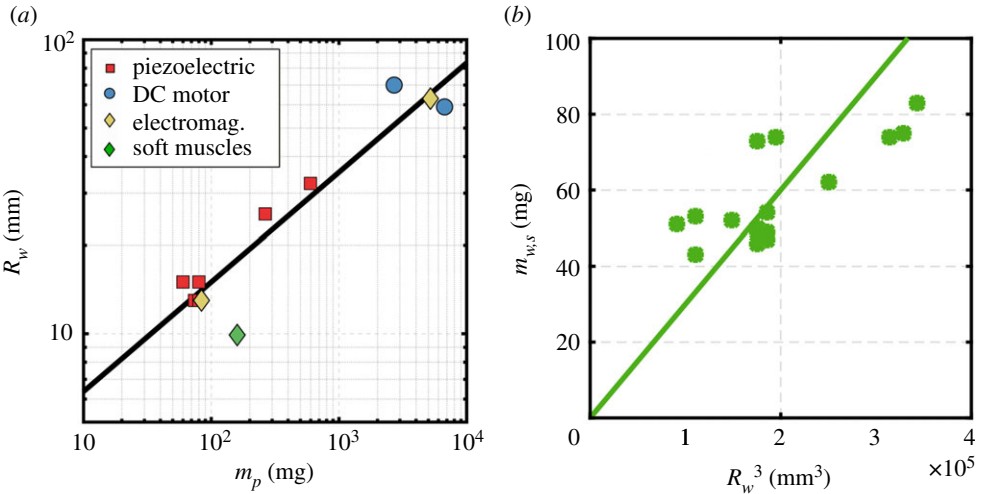

**Figure 2.** Scaling relations for resonant flapping wing concepts. (*a*) wing length ('split actuator' concept data point is not shown to avoid overcrowding); and (*b*) wing structural mass.

**Table 1.** Data for propulsion system mass and wing length of resonant robotic insect concepts with two wings.

| concept | actuation type | propulsion system mass, $m_P$ (mg) | wing length, $R_w$ (mm) |
|---|---|---|---|
| Harvard microrobotic fly [1] | piezoelectric | 60 | 15 |
| Harvard RoboBee [2] | piezoelectric | 80 | 15 |
| Harvard split actuator microrobotic bee [6,7] | piezoelectric | 70 | 15 |
| Harvard scaled robotic bee [8] | piezoelectric | 265 | 25.5 |
| Washington robotic fly [3] | piezoelectric | 74 | 13 |
| Shanghai Jiao Tong robotic insect-1 [4] | piezoelectric | 84 | 13 |
| Toyota robotic insect [5] | piezoelectric | 598 | 32.4 |
| Carnegie Mellon robotic insect [13] | DC motor | 2700 | 70 |
| Purdue robotic insect-1 [14–16] | DC motor | 6700 | 59 |
| Purdue robotic insect-2 [10] | electromagnetic | 5200 | 63 |
| Shanghai Jiao Tong robotic insect-2 [11] | electromagnetic | 80 | 13 |
| Harvard robotic insect-5 [12] | soft muscle | 160 | 9.9 |

complemented by the calculated standard error of regression estimate, $s_{y/x}$, which is a measure of the difference between actual values and values estimated from a given regression equation: the difference between actual data used to cast the relation and the corresponding estimated values from the regression scaling relation is calculated; the differences are then squared and summed; the sum is divided by the degrees of freedom and square root is taken leading to the value of $s_{y/x}$. Interestingly, the exponent of the wing length in equation (2.1) is slightly higher than 0.33, where 0.33 is the volume exponent if the scaling constant is the effective density. It is worth highlighting that the scaling laws developed throughout this study should always be considered as place holders within the design process and should always be updated whenever new relevant data becomes available.

The wing inertia is the other characteristic that is required within our modelling process. Here, we employ a simplified method to calculate the wing moment of inertia, $J_w$, based on the effective wing mass, $m_w$ and radius of gyration of the wing structure, $r_{mw}$

$$J_w = m_w r_{mw}^2, \tag{2.2}$$

where the effective wing mass can be expressed as

$$m_w = \left( m_{w,s} + \frac{m_{w,a} r_{mw,a}^2}{r_{mw}^2} \right). \tag{2.3}$$

Here $m_{w,a}$ is the wing aerodynamic inertia evaluated as $m_{w,a} = \rho(\pi/4)\bar{c}^2 R_w$, where $\rho$ is the air density, $\bar{c}$ is the mean geometric chord and $r_{mw,a}$ is

$$r_{mw,a}^2 = R_w^2 \int_0^1 (c(\hat{r})/\bar{c})^2 \hat{r}^2 \mathrm{d}\hat{r}; \quad \hat{r} = \frac{r}{R_w}. \tag{2.4}$$

We follow the common practice within the literature of neglecting the effect of wing aerodynamic inertia as in [26–28] since it is found to be much smaller compared with structural wing inertia. This simplification allows to obtain the effective wing mass directly from the mass of the wing structure itself, $m_{w,s}$. To obtain a scaling law for the wing structural mass we adopted the data in [26] which provide inertial characteristics for 16 candidates of wings used in insect-like flapping vehicles which are made of mylar film spanned over unidirectional carbon fibre frames. Figure 2b shows the wing structural mass data sourced from [26] together with our developed scaling law

$$m_{w,s}(\mathrm{mg}) = 3E^{-4} R_w^3(\mathrm{mm}^3), \quad n = 16, \quad R^2 = 0.95, \quad s_{m_{w,s}/R_w^3} = 14.3 \text{ mg}. \tag{2.5}$$

For simplicity, we evaluate the radius of gyration based on the assumption that the wing structural mass is uniformly distributed in the spanwise direction. As such, $r_{mw}$ can be expressed as a fraction of wing length, $R_w$

$$r_{mw} = \frac{R_w}{\sqrt{3}} = 0.577 R_w. \tag{2.6}$$

The radius of gyration of the 16 wing candidates used to cast equation (2.5) was measured by Roll *et al.* [26] based on the undamped compound pendulum formula. The obtained non-dimensional radius of gyration values ranged between 0.47 and 0.61 with an average value of 0.56 which compares remarkably well with the uniformly distributed mass assumption we adopt here.

## 2.2.2. Wing kinematics

For the purpose of our design and analysis framework, the wing is considered as a thin, rigid plate and actuated via a single rotational degree of freedom along the flapping axis. The wing has a second rotary degree of freedom in pitch along the wing spanwise axis. In the present model, it is assumed that this latter degree of freedom is passive (unactuated); however, is free to move in response to wing aerodynamic and inertial forces generated through the flapping motion. The dynamics of the pitch degree of freedom are not included within the overall system model. However, the underlying physics is captured by constraining the pitching motion kinematics to that possible with a passive hinge. It is understood that the structural dynamics of biological wings, such as those of real insects, can significantly affect the wing bending and twisting and hence the angle of attack distribution and its corresponding aerodynamic characteristics. On the other hand, wings of robotic insects (realized at different scales) are typically made of a high-strength membrane film spanning a stiffening structure made of carbon rods. These wings are always reported to have excellent strength-to-weight ratios and hence the assumption that the wings are rigid is deemed acceptable.

Here, we assume symmetric normal hovering flight with the wings moving along the horizontal stroke plane. This implies symmetrical forward and backward half-strokes and no deviation from a horizontal stroke plane. While these assumptions are mainly adopted for simplification, it is well established that these are reasonable approximations that adequately represent flapping flight of real insects together with ensuring reasonable levels of efficiency from an aerodynamic point of view [29,30]. The flapping angle time variation, $\phi(t)$, is considered based on a sinusoidal waveform which is a realistic representation of typical flapping variations within real and/or robotic insects. In fact, previous investigations of resonant motor-driven flapping vehicles have clearly demonstrated the accuracy of the adopted sinusoidal variation in resembling actual measured flapping angle variations of these systems [15,17]. As for the pitching kinematics, we demonstrate our results for both constant and sinusoidal waveforms of the geometric angle of attack, $\alpha_g(t)$, figure 3. These two types of waveforms represent the two extreme cases and hence allow for a convenient assessment of the pitching kinematics effect. Note that a sinusoidal variation provides a more realistic representation of the angle of attack waveform but introduces more complicated aerodynamic modelling expressions, as will be shown later. On the other

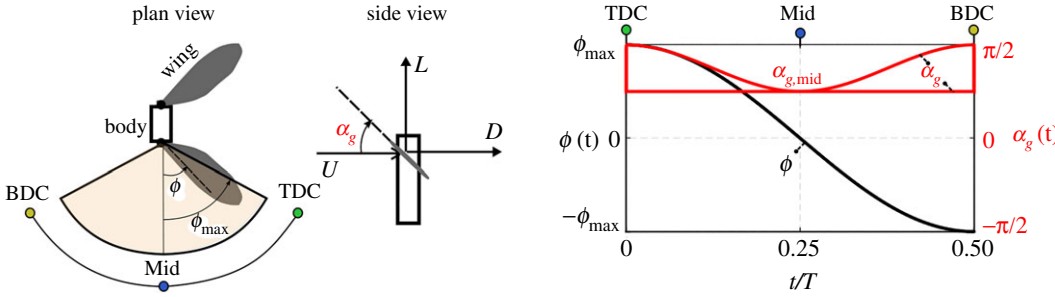

**Figure 3.** Wing kinematics adopted in this work. Owing to the symmetry of half-strokes, variations over only one half-stroke are shown. Flapping angle variation is sinusoidal. Both constant and sinusoidal angle of attack variations are shown. In reality, values of $\alpha_{g,\text{mid}}$ will be set by physical end-stops within the wing passive pitch mechanism. TDC denotes 'top dead centre', BDC denotes 'bottom dead centre' and Mid denotes 'mid half-stroke'.

hand, the constant angle of attack representation is still useful to consider as it is known to be the most effective pitching kinematic waveform in terms of lift generation [30]; it also allows for mathematical simplicity within models. As discussed in the Introduction, all previous modelling efforts (e.g. [13–18]) have only adopted a constant angle of attack representation.

### 2.2.3. Wing aerodynamics

The aerodynamic relations presented in this section only consider hovering flight as this flight mode is considered the main driver for propulsion system sizing. Hovering also provides relatively simple expressions for the lift and aerodynamic power owing to the absence of the forward speed component. Additionally, several forward flight performance characteristics, such as the maximum speed and range, can be estimated once the hovering requirements are identified [23,27]. Here, we adopt a quasi-steady treatment for aerodynamics allowing balance between accuracy and simplicity. The instantaneous lift and drag forces, $L$ and $D$, on a single flapping wing are given by

$$L = \frac{1}{2}\rho(\dot{\phi}(t)\hat{r}_2 R_w)^2 \left(\frac{R_w^2}{\text{AR}}\right) C_L(\alpha_g(t)) \tag{2.7}$$

and

$$D = \frac{1}{2}\rho(\dot{\phi}(t)\hat{r}_2 R_w)^2 \left(\frac{R_w^2}{\text{AR}}\right) C_D(\alpha_g(t)), \tag{2.8}$$

where $\hat{r}_2$ is the non-dimensional radius for second moment of wing area. Note that the non-dimensional radius for third moment of wing area, $\hat{r}_3$, will also be required within the modelling of the aerodynamic power. Hence, following biological inspiration, values for $\hat{r}_2$ and $\hat{r}_3$ against wing aspect ratio can be found using available data for real insects collected in [22]. Figure 4 shows the data sourced from [22] for the non-dimensional radius for first, second and third moment of wing area. Here, the use of data of real insects is deemed fit for purpose within our scope as most robotic insect wing planform shapes are bioinspired as well. From figure 4, it could be seen that for our selected wing aspect ratio of 3.5, $\hat{r}_2$ and $\hat{r}_3$ take the values of 0.54 and 0.59, respectively.

The lift and drag coefficients, $C_L$ and $C_D$, are defined in terms of the geometric angle of attack, $\alpha_g(t)$, based on the established nonlinear expressions [31,32]

$$C_L(\alpha_g(t)) = C_{L\alpha} \sin(\alpha_g(t))\cos(\alpha_g(t)) \tag{2.9}$$

and

$$C_D(\alpha_g(t)) = C_L(\alpha_g(t)) \tan(\alpha_g(t)), \tag{2.10}$$

where $C_{L\alpha}$ is the lift curve slope, which in general depends on the wing shape and Reynolds number. Note that equation (2.10) ignores the skin friction drag contribution (typically included as a constant value representing the drag coefficient at zero-lift angle of attack) which is adequate when considering

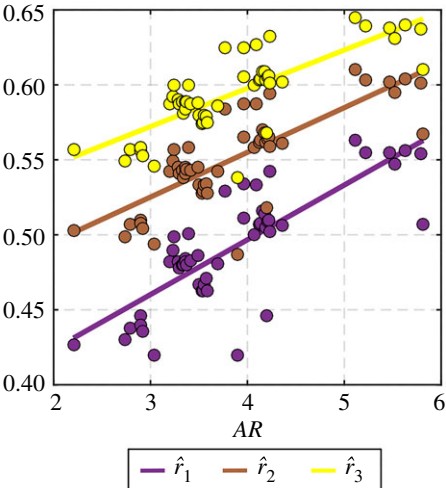

**Figure 4.** Scaling of non-dimensional radius of first, second and third moment of wing area against wing aspect ratio for real insects.

wings operating at high values of angle of attack [30]. The value of $C_{L\alpha}$ is evaluated based on the lifting line expression developed in [32]:

$$C_{L\alpha} = \frac{C_{l\alpha,2d}}{E + (k_{\text{ind}}k_{\text{tip}}k_{\text{flap}}C_{l\alpha,2d}/\pi\text{AR})}. \tag{2.11}$$

The two-dimensional aerofoil lift curve slope, $C_{l\alpha,2d}$, takes a value of 5.16 rad$^{-1}$ for flat plate wings at typical Reynolds numbers for insects [33]. The parameter $E$ is the quotient of the wing semi-perimeter to its length, and is included to correct the lifting line expression for low aspect ratio effects [34]. For a wing represented by a beta distribution with aspect ratio of 3.5 and $\hat{r}_2$ value of 0.54, $E$ takes the value 1.14 [34]. The parameters $k_{\text{ind}}$, $k_{\text{tip}}$ and $k_{\text{flap}}$ are the different contributors to the induced power factor included to correct for the difference in aerodynamic efficiency between assumed ideal uniform downwash distribution and real downwash distribution, and are evaluated based on the method proposed in [35]. $k_{\text{ind}}$ accounts for the non-uniformities of the downwash; it is a function of the wing planform geometry, and for typical insect-like wing planforms its value ranges between 1.1 and 1.3. For the wing planform considered here, it takes a value of 1.2 [35]. $k_{\text{tip}}$ accounts for the wake periodicity effect and has been shown to be well presented by a value of 1.1 [35]. $k_{\text{flap}}$ accounts for the reduction of the actuator disc area for flapping angle amplitudes below 90°. It is, thus, defined as

$$k_{\text{flap}} = \sqrt{\frac{\pi}{2\phi_{\text{max}}}}. \tag{2.12}$$

Clearly, as $\phi_{\text{max}}$ approaches 90°, the flapping actuator disc area reaches its maximum value. Considering the Rankine–Froude momentum theory, the increase in the actuator disc area to its maximum value leads the induced downwash velocity and consequently the induced aerodynamic power expended while producing a given amount of lift to achieve their minimum values [19,35]. This is reflected in equation (2.12) as $k_{\text{flap}}$ approaches unity (ideal value) when $\phi_{\text{max}}$ approaches 90°. Figure 5 shows how $C_{L\alpha}$ can change against different values of $\phi_{\text{max}}$ due to change in $k_{\text{flap}}$. It could be seen that a significant decrease in the flapping angle amplitude will reduce $k_{\text{flap}}$ and hence significantly deteriorates the aerodynamic coefficient values. As such, based on the previous, in this work a $\phi_{\text{max}}$ value of 90° will be adopted to ensure maximum *aerodynamic* efficiency.

It is worth mentioning that the Reynolds number, $Re$, will have a minor/negligible influence on the used aerodynamic coefficient relations within the range $1000 \leq Re \leq 15\,000$ [36]. This is the range that the scaling assessment presented in this study acts within. At Reynolds numbers higher than this range, flows usually get dominated by turbulent mixing, destabilizing the leading-edge vortex and limiting the maximum attainable lift coefficient values [37,38]. On the other hand, at Reynolds numbers lower than this range, viscous effects become dominant and their influence cannot be neglected [30,39].

Once the aerodynamic coefficients are evaluated, analytical expressions for the mean lift and aerodynamic power of the propulsion system can be obtained. The expressions for lift and aerodynamic

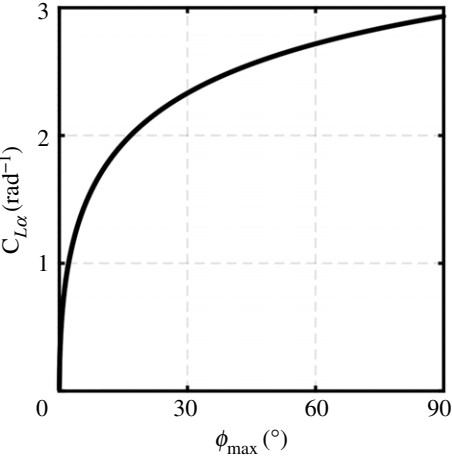

**Figure 5.** Variation of the wing lift curve slope with flapping angle amplitude.

power presented here are based on the first author's previous work in [30], hence only final expressions will be presented. For the constant angle of attack waveform, the mean lift is expressed as

$$L = \rho(R_w^2 \hat{r}_2^2)\left(\frac{2R_w^2}{\text{AR}}\right)(C_{L\alpha}\sin(\alpha_{g,\text{mid}})\cos(\alpha_{g,\text{mid}}))(\phi_{\text{max}}^2 \pi^2 f^2). \tag{2.13}$$

Using the above equation, the required flapping frequency, $f$, can be calculated to satisfy the propulsion system weight requirement ($W = m_p\, g$) as

$$f = \sqrt{\frac{16W}{\rho\pi^2 \underbrace{(4\phi_{\text{max}} R_w \hat{r}_2)^2}_{\propto U_{\text{mean}}^2} \underbrace{(2R_w^2/\text{AR})}_{S} \underbrace{(C_{L\alpha}\sin(\alpha_{g,\text{mid}})\cos(\alpha_{g,\text{mid}}))}_{C_L}}}. \tag{2.14}$$

The aerodynamic power (expended by one of the two flapping wings) can then be evaluated from

$$P_{\text{aero}} = 1.2W^{3/2}\sqrt{\frac{2}{\rho S}\frac{C_D}{C_L^{3/2}}}\left(\frac{\hat{r}_3}{\hat{r}_2}\right)^3 \times \frac{1}{2}. \tag{2.15}$$

As for the sinusoidal angle of attack waveform, the corresponding mean lift, frequency and aerodynamic power relations take the form

$$L = \rho f(R_w^2 \hat{r}_2^2)(2R_w^2/\text{AR})\left(\frac{C_{L\alpha}}{2}\right)\left(\frac{16}{3}\pi f\phi_{\text{max}}^2 \frac{\pi}{4}\left(\text{Hypergeom}\left[\{2\},\left\{\frac{3}{2},\frac{5}{2}\right\}, -\theta_{\text{mid}}^2\right]\right)\right), \tag{2.16}$$

$$f = \sqrt{\frac{3W}{\rho(4\phi_{\text{max}}R_w\hat{r}_2)^2(2R_w^2/\text{AR})\left(\frac{C_{L\alpha}}{2}\right)\left(\pi\theta_{\text{mid}}\left(\text{Hypergeom}\left[\{2\},\left\{\frac{3}{2},\frac{5}{2}\right\}, -\theta_{\text{mid}}^2\right]\right)\right)}}, \tag{2.17}$$

$$P_{\text{aero}} = W^{\frac{3}{2}}\sqrt{\frac{2}{\rho S}}\left(\frac{\hat{r}_3}{\hat{r}_2}\right)^3 \sqrt{\frac{3\pi(1 + \text{Hypergeom}[\{2\},\{1/2,5/2\}, -\theta_{\text{mid}}^2])^2}{32\left(\frac{C_{L\alpha}}{2}\right)\theta_{\text{mid}}^3(\text{Hypergeom}[\{2\},\{3/2,5/2\}, -\theta_{\text{mid}}^2])^3}} \times \frac{1}{2}, \tag{2.18}$$

where $\theta_{\text{mid}} = (\pi/2) - \alpha_{g,\text{mid}}$ and Hypergeom denotes the hypergeometric function. In this work, a mid half-stroke angle of attack of 45° will be used within all aerodynamic relations as this allows maximum lift coefficient as per equation (2.9).

## 2.3. Transmission mechanism considerations

For the class of vehicles considered in this study, gearhead transmissions are used with DC motors to modulate the motor speed and torque. Small-scale DC motors suitable for robotic insects (e.g. diameters in the range of 3–12 mm) typically operate at very high speeds (10 000–30 000 r.p.m.) and, therefore, a gearhead is required to reduce this speed. Planetary gearheads with transmission ratios ranging from 4 to 4000 are available from motor manufacturers and are usually ready for direct integration to the shaft of the motor.

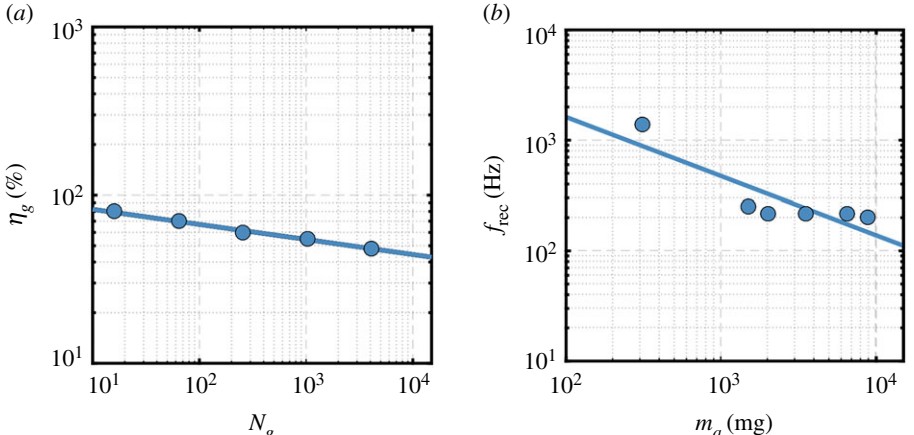

**Figure 6.** (a) Variation of gearhead efficiency against gearhead transmission ratio. (b) Variation of maximum operational recommended frequency against motor mass.

Within our developed framework, we require the evaluation of how two characteristics of gearheads scale. These are the gearhead transmission ratio, $N_g$, and gearhead efficiency, $\eta_g$. To build the relevant scaling relations, we have used the gearhead specifications from the motor manufacturer Faulhaber. Note that all scaling laws developed within the current work rely on relevant data collected from Faulhaber [40]. This is because it is the only available source that provides *complete* information on all characteristics needed over a range of motor sizes that are suitable for the scales we are interested in and hence is solely used throughout this work to ensure consistency. Figure 6a shows gearhead efficiency plotted against the transmission ratio for relevant gearheads (based on gearhead series 6/1, 8/1 and 10/1 from Faulhaber [40]). It could be seen that the gearhead efficiency decreases in a perfect exponential fashion with the increase of transmission ratio, and a function is fitted to the data providing the following scaling law:

$$\eta_g(\%) = 100N_g^{-0.09}, \quad n = 7, \quad R^2 = 0.99, \quad s_{\eta_g/N_g} = 1.5\%. \tag{2.19}$$

Values of flapping angular speed are expected to be lower than the operational angular speeds for a given DC motor. As such, a high gear transmission ratio will be required. Note that as the transmission ratio is increased, more gears are used to modulate the shaft rotation and this, in turn, increases the losses associated with the increased number of moving parts/frictional surfaces. Also, it is worth considering that as the transmission ratios are increased, additional gears are required, and this results in increased length and mass of the gearhead. The required gearhead transmission ratio is evaluated as the ratio of the maximum recommended operating speed of the motor, $\Omega_{rec}$, to the wing speed amplitude, $\Omega_{wing}$, evaluated as

$$N_g = \frac{\Omega_{rec}}{\Omega_{wing}} = \frac{2\pi f_{rec}}{2\pi f \phi_{max}}, \tag{2.20}$$

where $f_{rec}$ is the maximum recommended operational frequency for a given DC motor, $f$ is the required flapping frequency as evaluated by either equation (2.14) or (2.17), and $\phi_{max}$ is the maximum flapping amplitude. Note that by choosing the wing speed *amplitude* to calculate the transmission ratio, it is ensured that the motor is at least operating at its maximum recommended operating speed as recommended in [17]. Figure 6b shows how $f_{rec}$ scales with actuator mass, $m_a$, for relevant motor series: 0206, 0308, 0515, 0615, 0816, 1016 and 1024 from Faulhaber [40], from which a fitting relation is produced as

$$f_{rec}(Hz) = 19034m_a^{-0.535}(mg), \quad n = 7, \quad R^2 = 0.87, \quad s_{f_{rec}/m_a} = 245\,Hz. \tag{2.21}$$

Note that equation (2.20) assumes that the calculated gear ratio could always be realized from available gearheads. This is deemed acceptable for the purposes of our current scalability analysis; however, in practice, a designer may be forced to pick a specific gear ratio from the available gearhead options.

## 2.4. Actuator scaling laws

To assess scalability, we need to construct scaling laws for a number of actuator-related quantities that appear within the dynamics equation of motion that will be discussed in the next section. Note that, here, actuator refers to the DC motor. These quantities are: (i) moment of inertia of motor rotating

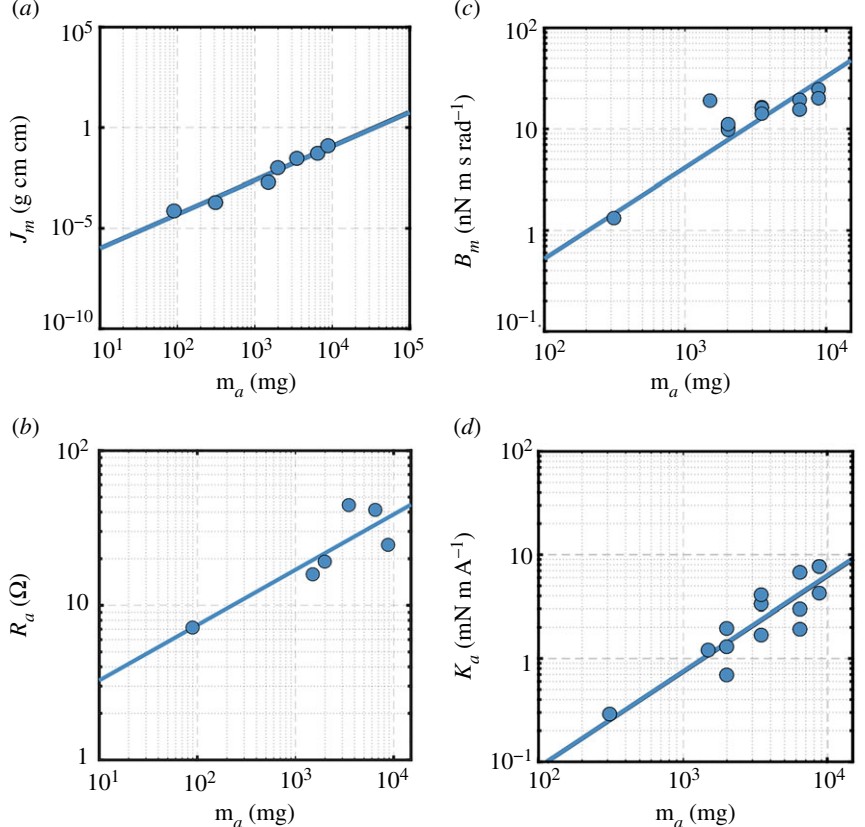

**Figure 7.** Scaling laws required for motor modelling. (*a*) Moment of inertia of motor rotating elements; (*b*) armature resistance; (*c*) damping coefficient of motor rotating elements and (*d*) motor torque constant.

elements, $J_m$; (ii) armature resistance, $R_a$; (iii) damping coefficient of motor rotating elements, $B_m$; and (iv) motor torque constant, $K_a$. Note that, here, we follow the actuator symbol notation used in [15]. The scaling laws were constructed based on data for relevant motor series: 0206, 0308, 0515, 0615, 0816, 1016 and 1024 from Faulhaber [40]. These data are plotted in figure 7, and the developed scaling laws are given as

$$J_m(\text{g cm}^2) = 2 \times 10^{-8} m_a^{1.6867}(\text{mg}), \quad n = 7, \quad R^2 = 0.97, \quad s_{J_m/m_a} = 0.015 \text{ g cm}^2, \tag{2.22}$$

$$R_a(\Omega) = 1.4335 m_a^{0.3578}(\text{mg}), \quad n = 6, \quad R^2 = 0.76, \quad s_{R_a/m_a} = 11.9 \, \Omega, \tag{2.23}$$

$$B_m(\text{nN m s rad}^{-1}) = 0.0084 m_a^{0.8993}(\text{mg}), \quad n = 15, \quad R^2 = 0.87, s_{B_m/m_a} = 6.4 \text{ nN m s rad}^{-1} \tag{2.24}$$

and $\quad K_a(\text{mN m A}^{-1}) = 0.0012 m_a^{0.9258}(\text{mg}), \quad n = 15, \quad R^2 = 0.89, \quad s_{K_a/m_a} = 1.5 \text{ mN m A}^{-1}. \tag{2.25}$

## 2.5. System dynamics and efficiency

Figure 8 shows a schematic diagram of the system components with their corresponding representations—the figure is a re-adaptation of the system diagram as depicted in [15]. This system is operating at resonance and subjected to nonlinear aerodynamic damping, hence is represented using a damped second-order lumped element model with an equation of motion of [15]:

$$J_s\ddot{\phi} + B_{s1}\dot{\phi} + B_{s2}|\dot{\phi}|\dot{\phi} + K_s\phi = K_v V(t). \tag{2.26}$$

Equation (2.26) is solved to evaluate the value of the AC voltage amplitude that would allow a sinusoidal flapping angle variation with an amplitude of 90°. The different terms in equation (2.26) are fully explained in table 2.

Once the system dynamics equation is solved for voltage amplitude that would ensure lift-off, it is instructive to estimate the system efficiency, defined as the ratio of the mean aerodynamic power expended by a single wing to produce sufficient lift, $P_{\text{aero}}$, to the mean required power to operate the motor driving this wing, $P_{\text{req}}$. The relations for the aerodynamic power expended by one wing were

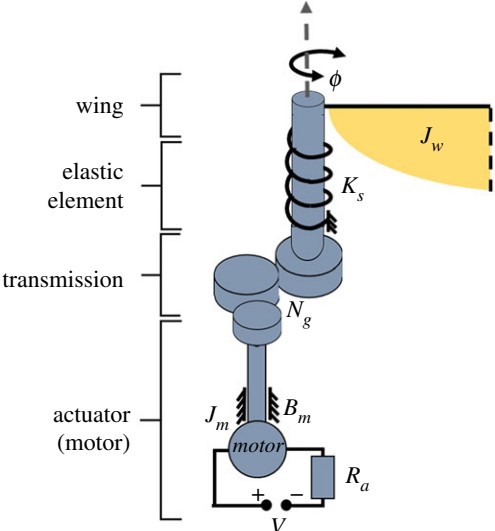

**Figure 8.** Schematic diagram for the modelling representation of the main system components. For compactness, only a part of the wing is shown.

**Table 2.** Variables of the system response equation including notation, equations and physical significance. Symbols in bold are these appearing in equation (2.26).

| symbol | description | evaluation |
|---|---|---|
| $J_s$ | total moment of inertia of all rotating components [15] | $J_s = \eta_g N_g^2 J_m + J_w + J_g$ |
| $J_g$ | gears moment of inertia | neglected—significantly lower compared with the other terms |
| $J_m$ | moment of inertia of motor rotating elements | scaling law—equation (2.22) |
| $J_w$ | wing moment of inertia | equation (2.2) |
| $\eta_g$ | gearhead efficiency | scaling law—equation (2.19) |
| $N_g$ | gearhead transmission ratio | scaling law—equation (2.20) |
| $B_{s1}$ | effective damping coefficient [15] | $B_{s1} = \eta_g N_g^2 \left( B_m + \dfrac{K_a^2}{R_a} \right)$ |
| $B_m$ | damping coefficient of motor rotating elements | scaling law—equation (2.24) |
| $K_a$ | motor torque constant | scaling law—equation (2.25) |
| $R_a$ | armature resistance | scaling law—equation (2.23) |
| $B_{s2}$ | aerodynamic damping coefficient [15] | $B_{s2} = \dfrac{1}{2}\rho \dfrac{R_w}{AR} \hat{r}_3^3 R_w^A C_D(\alpha_g(t))$ |
| $C_D(\alpha_g(t))$ | drag coefficient | equation (2.10) |
| $K_s$ | elastic element stiffness | $K_s = J_s(2\pi f)^2$ |
| $K_v$ | input gain [15] | $K_v = \eta_g N_g \dfrac{K_a}{R_a}$ |
| $V(t)$ | sinusoidal system excitation | $V(t) = V_{in}\cos(2\pi f t + \beta)$ |
| $f$ | frequency | equations (2.14) and (2.17) |
| $V_{in}$ | input AC voltage amplitude | output |
| $\beta$ | phase angle | 90° for resonant operation |

presented in equations (2.15) and (2.18) for constant and sinusoidal angle of attack variations, respectively, whereas the total required power input to a motor can be evaluated as [15]

$$P_{\text{req}} = \frac{1}{T} \int_0^T V(t) I(t) \, dt = \frac{1}{T} \int_0^T V(t) \left( \frac{V(t)}{R_a} - \frac{K_a N_g \dot{\phi}}{R_a} \right) dt,$$

(2.27)

where $T$ is the flapping period, $T = 1/f$. Evaluating the integration leads to

$$P_{\text{req}} = \frac{1}{2} \left( \frac{V_{\text{in}}}{R_a} - N_g \frac{K_a}{R_a} \phi_{\max}(2\pi f) \right) V_{\text{in}}.$$

(2.28)

As such, the system efficiency can be evaluated as

$$\eta = \frac{P_{\text{aero}}}{P_{\text{req}}}$$

(2.29)

It is useful to assess how the proposed conceptual design framework predictions compare with some of the available data of current existing designs. Here, we consider the two designs that rely on resonance and directly drive the wings through DC motor actuation, i.e. the designs from Carnegie Mellon [13] and Purdue [15] Universities. In this assessment, we will input to our framework the main inputs of these designs and evaluate the power that the motor is expected to deliver, $P_{\text{req}}$. There are two main reasons for picking $P_{\text{req}}$ for assessment. First, in the absence of any efficiency figures for the two designs under consideration, $P_{\text{req}}$ represents the final output; hence, providing judgement on the efficacy of the whole process. Second, and more importantly, the value of the power that the motor is expected to deliver, $P_{\text{req}}$, is arguably the most important piece of information that a designer would wish to know at the start of the design process as it enables the selection of suitable motor candidates for the vehicle under development.

For the Carnegie Mellon design, the following values are used [13]: actuator mass, $m_a$, of 1 g; total lift measured, which is used to estimate the propulsion system mass, $m_p$, of 3.16 g (lift-to-weight ratio of 1.17); wing aspect ratio, AR, of approximately 3.5; maximum flapping angle amplitude, $\phi_{\max}$, of 70°; and angle of attack value at mid half-stroke, $\alpha_{g,\text{mid}}$, of 45°. A $P_{\text{req}}$ value of 0.427 W is predicted as the average from the constant and sinusoidal angle of attack waveforms, comparing reasonably well with the reported measured value of 0.4 W [13]. As for the Purdue design, we used the reported values in [15]: $m_a$ of 2.5 g; an equivalent $m_p$ of 6.7 g; AR of 4.18; $\phi_{\max}$ of 55°; and $\alpha_{g,\text{mid}}$ of 45°. A $P_{\text{req}}$ value of 1.23 W is predicted as the average from the constant and sinusoidal angle of attack waveforms, which is in good agreement with the reported measured value of approximately 1.2 W [15]. These comparisons show that the proposed design tool can reasonably inform the designer of the power the motors should deliver, enabling a guided selection from the possible motor models at the relevant scale.

## 2.6. Rotary wing system

It is instructive to compare the efficiency levels from the flapping system described above with an analogous rotary system. This comparison is useful in providing a *preliminary* assessment of the relative advantage of different flying modes (flapping versus rotary) at different scales. In evaluating the efficiency levels for rotary actuation, we assume that the same vehicle configuration will hold, i.e. the vehicle still has two motors and each motor is driving the exact same wing used in flapping, but now in a rotary fashion. This way, it is ensured that all other elements of the comparison are kept the same, and only wing motion is varied. However, from a configuration point of view, such rotary system is not realized in such way as rotary vehicles typically adopt a propeller-like wing and usually adopt more than two motors. That said, this does not detract from the usefulness of this comparison as the main aim here is to understand the system efficiency levels when the same actuator operates in flapping or rotary motion under the same conditions.

For constant rotational speed and constant angle of attack, i.e. rotary motion, the aerodynamic power expended by a single wing and the frequency expressions can be evaluated based on [30]:

$$P_{\text{aero,rot}} = W^{3/2} \sqrt{\frac{2}{\rho S}} \frac{C_D}{C_L^{3/2}} \left( \frac{\hat{r}_3}{\hat{r}_2} \right)^3 \times \frac{1}{2}$$

(2.30)

and

$$f_{\text{rot}} = \sqrt{\frac{2W}{\rho (2\pi R_w \hat{r}_2)^2 \, (2R_w^2/\text{AR}) \, (C_{L\alpha} \sin(\alpha_g) \cos(\alpha_g))}},$$

(2.31)

where $W$ is the weight of the whole system, $S$ is the area of both wings and $\alpha_g$ is the geometric angle of attack set as 45° for consistency. Note that both rotary (also so-called revolving) and flapping wing motions share the same expression for the lift coefficient [30,31,34]. Direct comparison of equation (2.30) with equation (2.15) shows that the rotary aerodynamic power expression is 20% less in value compared with sinusoidal flapping with constant angle of attack. This implies that rotary motion will expend less aerodynamic power to support the same weight, and thus will always be aerodynamically more efficient. Once motion frequency is evaluated using equation (2.31), values for gearhead transmission ratio, $N_{g,\mathrm{rot}}$, and gearhead efficiency, $\eta_{g,\mathrm{rot}}$, for the rotary system can be evaluated in a similar fashion to using equations (2.19)–(2.21).

The torque of the rotary system, $T_{\mathrm{rot}}$, at the motor output point can be evaluated from:

$$T_{\mathrm{rot}} = \frac{P_{\mathrm{aero,rot}}}{\eta_{g,\mathrm{rot}}\Omega_{\mathrm{rec}}}, \tag{2.32}$$

where $\Omega_{\mathrm{rec}}$ is the motor recommended rotation speed as defined in equations (2.20) and (2.21). The motor operating current can, thus, be obtained as

$$I_{\mathrm{rot}} = \frac{T_{\mathrm{rot}}}{K_a} + I_0, \tag{2.33}$$

where $I_0$ is the no-load current which for the series of relevant motors: 0206, 0308, 0615, 0816, 1016 and 1024 from Faulhaber [40], a fitting relation can be produced to represent it as

$$I_0\,(\mathrm{A}) = 0.1998 m_a^{-0.359}\,(\mathrm{mg}), \quad n = 6, \quad R^2 = 0.85, \quad s_{I_0/m_a} = 0.0054\,\mathrm{A}. \tag{2.34}$$

The motor supply voltage is, then, calculated as

$$V_{\mathrm{rot}} = R_a I_{\mathrm{rot}} + K_a \Omega_{\mathrm{rec}}. \tag{2.35}$$

The rotary system efficiency can, therefore, be obtained as

$$\eta = \frac{P_{\mathrm{aero,rot}}}{V_{\mathrm{rot}} I_{\mathrm{rot}}}. \tag{2.36}$$

# 3. Results

## 3.1. Solution procedure

Before addressing any of the outputs, it is useful to briefly explain how the different models are deployed and integrated. Figure 9 provides an overview of the whole solution procedure. There are two main inputs: the actuator mass, $m_a$, and the actuator to propulsion system mass ratio, $\mu_a$. Note that once the previous two inputs are defined, this implicitly assigns a value for the propulsion system mass. There are six main final outputs: flapping frequency required for lift-off, $f$, stiffness of the elastic element for resonant operation, $K_s$, required AC voltage amplitude, $V_{\mathrm{in}}$, aerodynamic power expended by a flapping wing, $P_{\mathrm{aero}}$, required supply input power to a motor, $P_{\mathrm{req}}$ and system efficiency, $\eta$.

The solution procedure starts by feeding the two inputs into the developed scaling laws to provide several intermediate outputs, namely actuator specifications (maximum recommended motor operational frequency, $f_{\mathrm{rec}}$, motor moment of inertia, $J_m$, armature resistance, $R_a$, damping coefficient of motor rotating elements, $B_m$, and motor torque constant, $K_a$) as well as the wing parameters (wing length, $R_w$, and wing structural mass, $m_{w,s}$). The wing parameters then allow to calculate the wing moment of inertia, $J_w$. Simultaneously, wing planform parameters, AR, $\hat{r}_2$ and $\hat{r}_3$, are defined as per the designer's choice (here an aspect ratio of 3.5 is used as discussed previously) and using these parameters together with the wing length one can fully define the wing planform shape and determine the wing area, $S$. Wing kinematics are defined based on sinusoidal flapping angle variation (of 90° amplitude) and wing pitching with either constant or sinusoidal angle of attack variation (of 45° amplitude), see figure 3.

All inputs and intermediate outputs are then used to evaluate the final outputs. However, since many constituting parameters are dependent on each other, the solution consists of two parts: the aerodynamic model and the system dynamics model. The former model uses the lift coefficient equation together with the wing planform geometric and kinematic characteristics to evaluate the required frequency that would allow a lift-to-weight ratio of unity. The aerodynamic model also evaluates the aerodynamic drag/

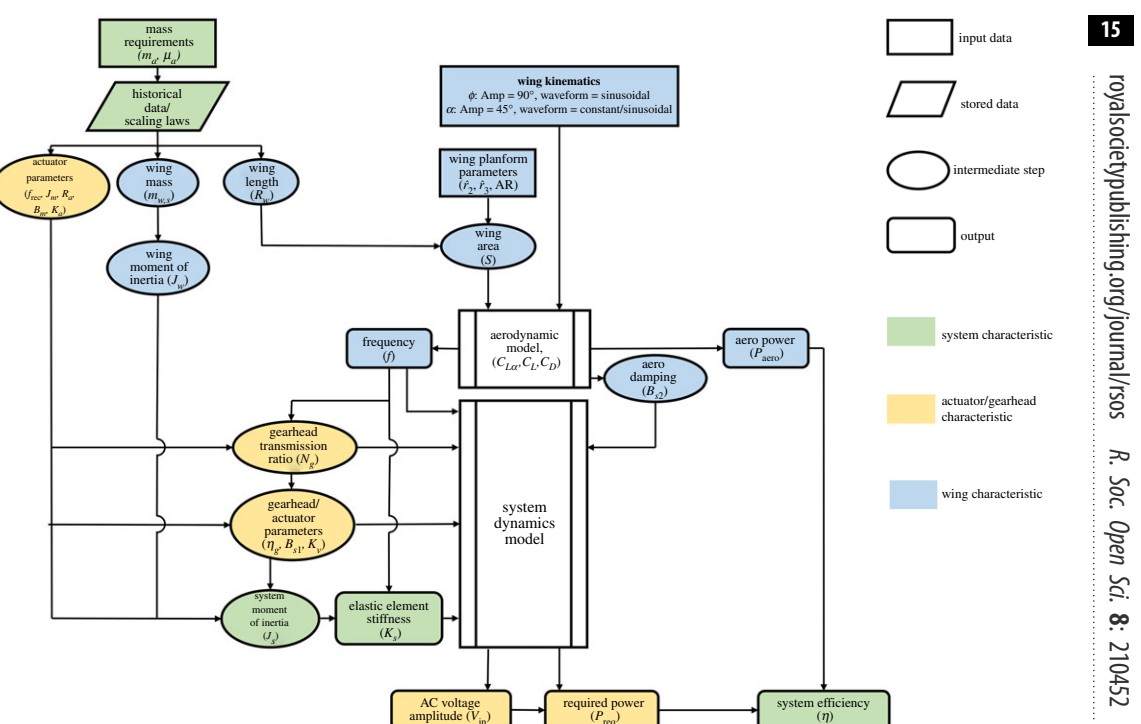

**Figure 9.** Summary diagram of the developed solution procedure.

damping term, $B_{s2}$, required within the system dynamics model, as well as the aerodynamic power needed to evaluate the system efficiency. It should be noted that the current aerodynamic model considers lift-off as its primary objective. Hence, the flapping angle amplitude is fixed to its maximum possible value to ensure highest *aerodynamic* efficiency, whereas only flapping frequency is evaluated to guarantee sufficient lift is generated to balance weight. In the absence of any known design constraints, this approach is deemed adequate (or at least sufficient); however, designers may opt to evaluate the flapping amplitude for a constrained frequency or evaluate both frequency and flapping amplitude to achieve a certain design objective. Moreover, the combined modulation of both flapping frequency and amplitude should be identified for 'off-design' conditions, e.g. in manoeuvring flight, to control the amount of aerodynamic force generated. Ultimately, optimum combinations of flapping frequency and amplitude for different flight conditions and their corresponding input electric signal parameters should be defined. These can then be programmed within the controller circuit to ensure that optimum parameters are always employed for the different flight scenarios.

The maximum recommended motor operational frequency initially determined using scaling laws is used with the obtained flapping frequency and the maximum flapping angle amplitude, as per equation (2.20), to define the required gearhead transmission ratio, $N_g$. This transmission ratio is then used in conjunction with the actuator parameters to define values for the gearhead efficiency, $\eta_g$, the effective damping coefficient, $B_{s1}$ and the input gain parameter, $K_v$ (also, see table 2). The wing inertia and the motor rotating parts inertia are used to define the system moment of inertia, $J_s$, and, subsequently, the elastic element stiffness can be calculated using the inertia and flapping frequency information. The dynamic behaviour is then considered by solving the system equation of motion (equation (2.26)). This allows to evaluate the required operating AC voltage amplitude which consequently allows for the evaluation of the required supply input power. Finally, the system efficiency is evaluated using the values of aerodynamic power and required supply input power.

## 3.2. Operational requirements and system performance

Outputs of the developed framework are shown in figures 10 and 11. Figure 10a-i demonstrates the flapping wing frequency (equivalent to the required system resonant frequency) as a function of the actuator mass for different actuator mass ratios for constant angle of attack kinematics. It is worth reiterating that the actuator mass is that of a single actuator of the two involved in the vehicle. Here, we considered actuator mass values spanning between 100 mg and 10 g providing a convenient range

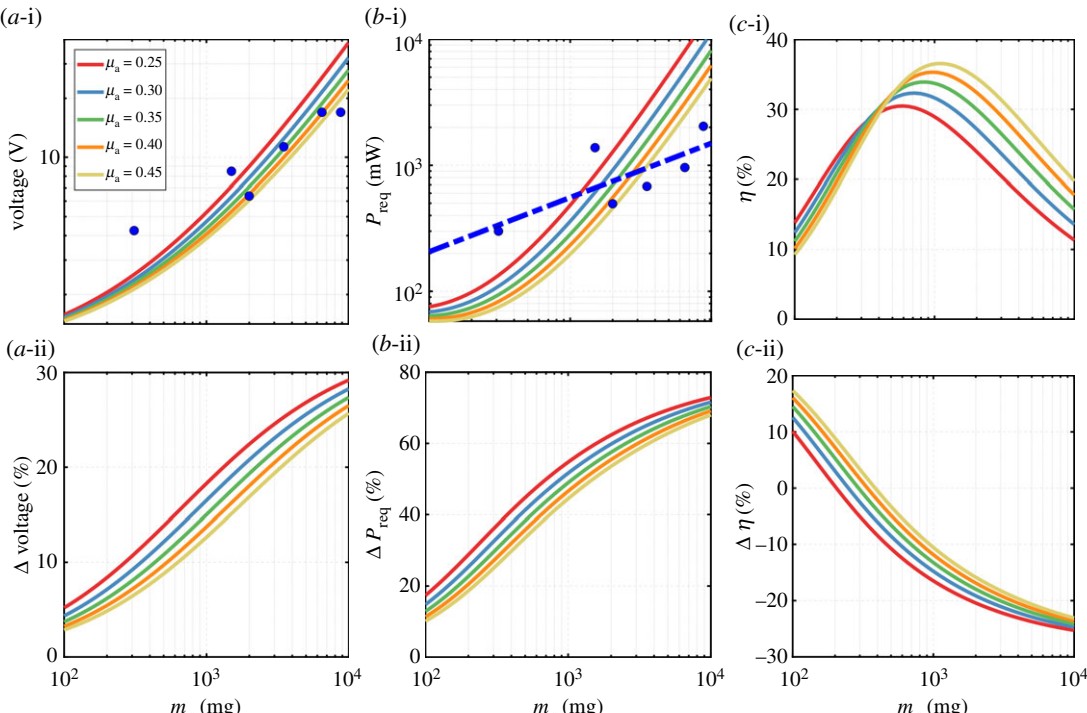

**Figure 10.** Variation of operational parameters against actuator mass for different actuator mass ratios. (*a*-i) frequency, (*b*-i) aerodynamic power and (*c*-i) elastic element stiffness variations for constant angle of attack kinematics. Percentage difference in values of (*a*-ii) frequency, (*b*-ii) aerodynamic power and (*c*-ii) elastic element stiffness when a sinusoidal angle of attack variation is adopted as opposed to constant angle of attack.

**Figure 11.** Variation of system performance parameters against actuator mass for different actuator mass ratios. (*a*-i) AC voltage amplitude, (*b*-i) required input power and (*c*-i) system efficiency variations for constant angle of attack kinematics. Percentage difference in values of (*a*-ii) AC voltage amplitude, (*b*-ii) required input power and (*c*-ii) system efficiency when a sinusoidal angle of attack variation is adopted as opposed to constant angle of attack. Markers in (*a*-i) indicate nominal voltages based on Faulhaber motors characteristics. Markers and trendline in (*b*-i) are for maximum operation power values based on the characteristics of Faulhaber motors.

to investigate scalability aspects for the class of air vehicles considered in this study. Additionally, this is the range of the data used to develop our scaling laws. The required flapping frequency decreased from approximately 40 to 13 Hz as the actuator mass increased from 100 mg to 10 g. As expected, frequency values decrease with increasing actuator mass values for a given actuator mass ratio; also, it is evident that an increase in the actuator mass ratio (equivalent to a decrease in the propulsion system mass for a given actuator mass) leads to an increase in flapping frequency. Nevertheless, the rate of increase in frequency decreases as the actuator mass ratio increases.

To better understand the effect of pitch kinematics on operation requirements, figure 10$a$-ii demonstrates the effect of employing sinusoidal kinematics on the wing-flapping frequency. Here, we show the results as the percentage difference in frequency when adopting sinusoidal angle of attack variation compared with the case where constant angle of attack variation is employed. Note that percentage differences in quantities presented in this section are evaluated as the difference in the quantity value when adopting sinusoidal and constant angle of attack variations, respectively divided by the quantity value when adopting constant angle of attack variation. With sinusoidal angle of attack variation, one can observe an increase in the wing-flapping frequency of 3.2%; however, this increase is independent of the actuator mass and/or the actuator mass ratio. This rise in flapping frequency is expected as the wing will now operate for a longer duration away from the optimum 45° angle of attack (figure 3) that is known to maximize the lift coefficient. Therefore, to achieve the same lift force production, the frequency needs to increase to compensate for the decrease in the coefficient value.

The second aerodynamic output considered here is the aerodynamic power expended by a flapping wing and is illustrated in figure 10$b$-i for the constant angle of attack case. It is evident that for a given actuator mass ratio as the actuator mass increases (and hence propulsion system mass), power expended against aerodynamic drag experiences a significant rise. This is expected as the aerodynamic power is known to increase considerably when the lift requirement is increased. Roughly, it can be seen that there is an order of magnitude increase in aerodynamic power as the actuator mass increases by an order of magnitude. On the other hand, figure 10$b$-ii shows that the adoption of sinusoidal angle of attack kinematics leads to an increase of 29% in the aerodynamic power expenditure. This increase is, again, independent of the actuator mass and/or actuator mass ratio.

Elastic element stiffness is yet another fundamental parameter for resonant system operation as it allows matching of the flapping frequency to the system natural frequency. Figure 10$c$-i shows the required elastic element stiffness values to ensure system resonance while adopting constant angle of attack kinematics. The stiffness values increase abruptly as the actuator mass increases and decrease as the actuator mass ratio increases. However, the sensitivity to the value of actuator mass ratio decreases as the actuator mass decreases. From figure 10$c$-ii, it is evident that adopting sinusoidal angle of attack kinematics will require an increase in the stiffness of the elastic element. By contrast to observations for frequency and aerodynamic power, this increase in stiffness value is a function of the actuator mass and the actuator mass ratio: an increase of approximately 1 to 6% in stiffness will be required as the actuator mass increases from 100 mg to 10 g. It is worth remembering that the only scaling law involved in the frequency and aerodynamic power results is that of the wing length. On the other hand, the stiffness results are based on scaling laws for the wing length and system inertia.

It is now instructive to look at the system operation requirements in terms of the required AC voltage amplitude for successful operation, required input supply power and subsequent system efficiency. Figure 11$a$-i demonstrates the operational voltage waveform amplitude, $V_{in}$, that would allow for sufficient lift generation to balance the propulsion system weight. Voltage values increase from approximately 1.5 V to almost 40 V for actuator mass values ranging from 100 mg to 10 g. For a given actuator mass, required voltage values decrease with an increase in actuator mass ratio; however, this decrease becomes less relevant at lower actuator mass values. Figure 11$a$-i also demonstrates markers representing nominal voltage values for relevant Faulhaber motor series used in this study [40]. Note that this is the nominal AC voltage amplitude evaluated by multiplying the recommended nominal DC value by a square root of two, following [13]. It is evident that most marker values lie within the evaluated voltage ranges, demonstrating that available motors are generally capable of satisfying the voltage requirement of the flapping wing propulsion system application. Figure 11$a$-ii shows the effect of adopting sinusoidal kinematics on the required voltage values and demonstrates a required increase of voltage ranging from approximately 4 up to 28% as the actuator mass increases.

Figure 11$b$-i shows the power required to be supplied to the motor to ensure a unity lift-to-weight ratio value. As expected, the required power behaviour is an increasing behaviour with the increase of actuator mass. To better understand how the obtained input power requirement compares with what available motors can actually produce, markers (and a fitting line) are added to figure 11$b$-i demonstrating the

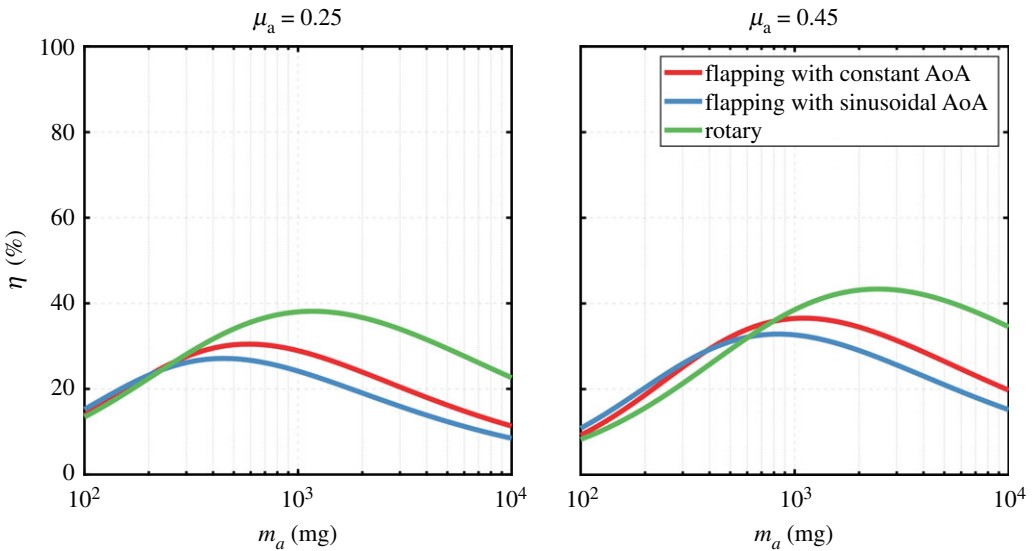

**Figure 12.** Comparison between efficiency levels of flapping and rotary wing systems. AoA denotes the angle of attack waveform.

maximum operation power values for relevant motors series from Faulhaber. In fact, the values of these markers are evaluated based on the thermal limits of the maximum permissible standard rotor temperature [40]. Judging by the obtained trendline for power data, it is interesting to find that available motors are able to satisfy the required power demand only for an actuator mass below roughly 2 g. Finally, figure 11$b$-ii shows the effect of adopting sinusoidal kinematics which leads to an increase in the required power values by 13 to 70% over the range of actuator masses considered.

Figure 11$c$-i illustrates the system efficiency values for constant angle of attack kinematics. The general trend is that the efficiency lines will increase and then decrease leading to a clear global maximum (peak) value. Peak efficiency values range from 30 to 36% giving an indication of the efficiency level expected from these systems. Interestingly, the peak efficiency value will change as the actuator mass ratio varies in such a way that as the actuator mass ratio increases, the peak efficiency will increase and will occur at a higher actuator mass value. Remarkably, the use of sinusoidal angle of attack kinematics (figure 11$c$-ii) can lead to an increase in efficiency. This is striking given that the use of sinusoidal angle of attack kinematics requires higher values for both aerodynamic and supply power values. Nevertheless, the quotient of these two higher power values can lead to better efficiency performance, and this becomes evident as the actuator mass decreases.

Finally, figure 12 compares the efficiency levels from the flapping wing system considered in this work with that obtained from the conceived rotary wing system presented in §2.6. In this comparison, we show results for $\mu_a$ values representing the bounds of the range considered in this study. It is evident that increasing $\mu_a$ increases the system efficiency values and moves the efficiency peaks towards higher actuator mass values. However, $\mu_a$ has no influence on the general trends of the system efficiency curves, hence does not influence the main conclusion from this demonstration. The main outcome of this demonstration is that at small scales, the system efficiency levels of the flapping system described in this work are slightly higher than that when a rotary system is adopted. In fact, the efficiency values of the rotary system deteriorate at small scales and this is mainly due to the higher no-load current, $I_0$, values as the motor scale decreases. On the other hand, as scale increases, the rotary system efficiency levels increase, significantly surpassing the flapping system efficiency levels. This result is in line with the findings of Hawkes & Lentink [41] who showed that flapping wing vehicles actuated with piezoelectric actuators have higher system efficiency when compared with rotary motor-actuated vehicles at small scales similar to tiny insects and this trend flips as scale increases.

## 4. Discussion

The development of flapping wing vehicles requires generic toolsets that would allow conceptual design/sizing of such vehicles while taking into consideration the correct interaction between the different subsystems in a transparent fashion. Design methods have been previously presented for

different flapping wing vehicle configurations, including design tools for concepts developed for forward [42,43] and hovering [44] flight modes. Recent progress in miniaturization of engineering systems together with the desire to create tiny aerial robots have motivated the development of a range of flapping wing vehicle concepts at insect-scale. While the flapping wing motions from these concepts are generally similar, the approach through which the flapping motion is created differs between concepts, owing to the different actuators and motion amplification mechanisms adopted. In this work, we presented a framework for the conceptual design of micro-DC motor-actuated insect-like flapping wing vehicles that exploit resonance as a means for successful operation. A 'multi-physics' modelling approach linking the aerodynamics, system dynamics and electrical domains is presented allowing designers to explore interactions between domains in a single design environment. The work specifically considered scalability aspects of this class of air vehicles and provided preliminary scaling laws to describe how characteristics of wings, gearhead transmissions and micro-DC motors scale with actuator mass. This allowed the developed framework to facilitate intuitive investigation of the available design space over the range of relevant actuator mass values.

The current work considered quasi-steady treatments for the wing aerodynamics and adopted simple but realistic wing kinematics, allowing the solution procedure to begin with an explicit evaluation of the flapping frequency for weight support based on the supplied wing kinematics and geometric data. An important feature of the current air vehicle design problem is the aero-mechanical coupling that leads to unique aspects of the design problem: the design process needs to find the operational flapping frequency at the start which is similar to finding the head speed in rotary vehicles design. However, because the system is resonant, actuator physical sizing comes into play. Also, a fixed or a rotary wing vehicle design does not include acceleration considerations; however, for a flapping wing, inertial loads become part of the design of the propulsion system. At the heart of the process lies the system dynamics simulation which interfaces the aerodynamics to the electro-mechanics and identifies the operating voltage amplitude to achieve the required wing motion as well as the input power to be supplied to the actuator. Information on the output aerodynamic power and input electrical power, subsequently, enables to evaluate the system efficiency.

The design process started with only two inputs: the actuator mass and the actuator mass ratio which both then define the whole propulsion system mass. The process outputs were a number of operation and performance metrics that enable a designer to evaluate the solution against higher level trades within the overall vehicle design loop. The developed conceptual design process was presented through conducting a scalability analysis on how these operation and performance metrics vary with the actuator mass (ranging from 100 mg to 10 g) and actuator mass ratio (ranging from 0.25 up to 0.45). Comparing against data of currently available motor operational characteristics, it was found that most motors operate at nominal voltages within the required values by this class of vehicles; however, a trendline developed from these data suggested that only motors with mass less than 2 g are capable of satisfying the required input power demand. Nevertheless, this conclusion should be considered with caution as motor characteristics can change as technology in this area improves, particularly with the recent increased demand on actuators for microsystem applications.

A useful contribution of the current work is investigating the effect of adopting the more realistic sinusoidal angle of attack kinematics as opposed to the oversimplified constant angle of attack assumption usually adopted in the literature. It was found that sinusoidal angle of attack kinematics will always lead to an increased requirement when considering the metrics investigated in this study. For example, a 3.2% increase in flapping frequency, a 29% increase in aerodynamic power, a 1–6% increase in elastic element stiffness, a 4–28% increase in voltage and a 13–70% increase in the input power will be required. These values are significant and should be taken into account when considering design and/or off-design conditions. This is because a considerably different performance of the system will be obtained when considering this more demanding but also more realistic kinematics.

The developed framework also provided useful information on the optimum operation of this class of vehicles. It has been shown that when adopting constant angle of attack kinematics, a peak efficiency of 30% occurs at an actuator mass value of 0.5 g for an actuator mass ratio of 0.25 and this peak efficiency increases to a maximum of 36% for an actuator mass of 1 g with an actuator mass ratio of 0.45. Nevertheless, probably the most striking outcome is that the efficiency of the system can get better when adopting sinusoidal angle of attack kinematics. In fact, when adopting the sinusoidal angle of attack kinematics, an increase of approximately 15% in efficiency would happen for an actuator mass of 100 mg, and this increase in efficiency will decrease to approximately −25% at the upper bound actuator mass of 10 g.

Following the above discussions, it is informative to provide some design guidelines and suggestions for future designs:

(1) The design of an air vehicle would normally start by considering the basic geometric characteristics of its wings. For resonant flapping wing vehicles, the developed scaling law in equation (2.1) can provide a reasonable starting point to determine the wing scale. On the other hand, the selection of the wing aspect ratio is considered vital as it defines the severity of transformation from two-dimensional to three-dimensional wing aerodynamics. However, choices here are limited and only a narrow range of three to four is deemed practical. A wing with an aspect ratio higher than four mainly suffers from stall particularly at wing regions close to the tip, whereas wings with aspect ratios lower than three will experience reduced aerodynamic performance due to increased three-dimensional induced aerodynamic effects. Wing chord distribution can be considered at a later stage of the design process to tune the aerodynamics to desired performances (e.g. as proposed in [20]) depending on vehicle requirements.

(2) Following wing geometric considerations, wing kinematics should be decided. A sinusoidal variation of the wing-flapping motion is a practical choice for resonant flapping wing vehicles; however, the amplitude of the flapping angle depends on the actuator involved and the degree of sophistication of its associated transmission mechanism. For example, piezoelectric or electromagnetic actuated designs typically involve relatively sophisticated transmission mechanisms unable to allow the reach to a maximum flapping amplitude angle of 90°. However, resonant motor-driven vehicles, considered in this study, can practically achieve such a maximum amplitude value. This, in turn, allows the wing to scan a larger flapping disc area and hence improve the aerodynamic efficiency of lift production.

(3) Pitching kinematics are typically realized with passive wing hinges, and designers can opt to design these hinges either to enable higher lift coefficient values if a constant angle of attack of 45° can be maintained through the half-strokes or to allow possible higher system efficiency values if the hinge is designed to allow a sinusoidal variation of the angle of attack.

(4) The system elastic element should be selected, for given system inertia, to allow system resonance at a frequency that would enable sufficient wing speed to satisfy the lift production demand.

(5) The gearbox should be selected to allow an appropriate gear ratio between the wing speed and the recommended speed for motor operation in such a way that the latter speed is not exceeded under different operating conditions.

(6) Motor selection is primarily decided by the ability of a motor candidate to deliver the required power for the application, and current motors with masses up to 1–2 g seem to be able to deliver sufficient power for the resonant motor-driven flapping wing vehicle application.

(7) Finally, a resonant motor-driven flapping wing vehicle is a more efficient system at small scales; however, as scale increases, switching to a rotary wing system becomes significantly more efficient.

As a final comment, it should be noted that the main design objective within the proposed design process is based on the satisfaction of an effectiveness criterion (i.e. lift = weight) rather than efficiency. However, this is an acceptable limitation at the current state of technology and given that the primary goal at present is to achieve self-supported hovering flapping flight at insect scale. Another point to consider is that the developed design process, as any other conceptual design process, uses developed scaling laws based on currently available heuristic data. However, the level of uncertainty in technology estimates and uncertainty associated with manufacturing is currently high due to the lack of enough historical successful vehicles and actuator characteristics at this scale upon which estimation heuristics can be based. Therefore, it is necessary to note that the presented design process scaling laws should always be updated with reliable relevant data when available. This should be done while considering measures for assessing the variability in data used to produce design scaling laws, together with any potential hysteresis issues of the used subsystems and sensors.

Data accessibility. All required data are contained in the paper. The computation code for the main results shown in figures 10 and 11 is provided as electronic supplementary material.

Authors' contributions. M.R.A.N. conceived and designed the study, performed the design evaluations, analysed the data and wrote the paper; R.M. helped in design evaluations and writing and revising the manuscript. All authors gave their final approval for publication.

Competing interests. We declare we have no competing interests.

Funding. This work was funded by the Leverhulme Trust through Research Project grant no. RPG-2019-366.

Acknowledgements. We would like to acknowledge the combined contributions by Diven Chauhan, Clinton Jonga, Simas Kaminskas, Tanya Shaban, Dilan Shah, Joshua Tampoe-Holmes, Dehao Wang, Jie Wang and Wei Wu towards

developing the example resonant motor-driven propulsion system demonstrated in electronic supplementary material, video. The authors are also grateful to Mark Quinn for the valuable discussions on uncertainty analysis.

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
