## [Peer Review File · Royal Society Open Science]

Review History

RSOS-210452.R0 (Original submission)

Review form: Reviewer 1

Is the manuscript scientifically sound in its present form?

Yes

Are the interpretations and conclusions justified by the results?

Yes

Is the language acceptable?

Yes

Do you have any ethical concerns with this paper?

No

Have you any concerns about statistical analyses in this paper?

No

Recommendation?

Major revision is needed (please make suggestions in comments)

Comments to the Author(s)

This work performed a scaling analysis on a model flapping wing system. It calculated the dependency of system design parameters and performance (e.g., actuation voltage, wing frequency, power, efficiency) on the wing actuator mass. To enable this analysis, the authors obtained empirical scaling rules of electric motors, quasi-steady aerodynamic model, and other empirical scaling rules on the wing length, propulsion-system mass, as well as using different profiles of angle of attack trajectory.

I think this is an interesting subject, the paper is clearly written and easily to follow. However, to enhance the contribution and significance of the work, I suggest the following.

1. The model used is highly, highly simplified! We know that wing design/structural dynamics have significant effects on the AoA (twisting/bending), the lift generation, efficiency etc. And its scaling is highly complex and different in animals (different species of insects, hummingbirds, etc) and potentially in robotic systems as well, and we don't know if the rigid wing assumption applies well to all scales. Also the quasi-steady model also depend on the scales (e.g., Reynolds number, which has been modeled in the literature) in addition to aspect ratio. The wing actuation mechanisms also varies from direct-motor drive with gears, to those with four-bar, to piezoelectric and electromagnetic. The current model based only on the direct motor drive model. Considering all these complexities, I'm really skeptical that the conclusions derived based on this model could have strong practical significance. Therefore, to enhance this aspect, I would suggest 1) authors to collect the actual data on the existing designs of flapping wing robots and integrate them with the predicted performance of the system, 2) based on the above comparison/analyses, provide some guidelines or at least some suggestions for future designs.
2. I would also suggest compare with rotary wing system, to my understanding, the rotary wing system should be more efficiency in hovering flight than the flapping wing system, at least in medium to large scales. This is important to understand at which scale the flapping wing flight becomes more advantageous.
3. Please show the source of data in Fig. 2 and 4.

Review form: Reviewer 2 (Moatasem Fouda)

Is the manuscript scientifically sound in its present form?

Yes

Are the interpretations and conclusions justified by the results?

Yes

Is the language acceptable?

Yes

Do you have any ethical concerns with this paper?

No

Have you any concerns about statistical analyses in this paper?

No

Recommendation?

Major revision is needed (please make suggestions in comments)

Comments to the Author(s)

Please refer to the attached document (see Appendix A).

Review form: Reviewer 3 (Mohamed Zakaria)**Is the manuscript scientifically sound in its present form?**

Yes

Are the interpretations and conclusions justified by the results?

Yes

Is the language acceptable?

Yes

Do you have any ethical concerns with this paper?

No

Have you any concerns about statistical analyses in this paper?

No

Recommendation?

Accept with minor revision (please list in comments)

Comments to the Author(s)

Thanks for the effort done in this manuscript (see Appendix B).

Decision letter (RSOS-210452.R0)

Dear Dr Nabawy,

The Editors assigned to your paper RSOS-210452 "Scalability of Resonant Motor-Driven Flapping Wing Propulsion Systems" have now received comments from reviewers and would like you to revise the paper in accordance with the reviewer comments and any comments from the Editors. Please note this decision does not guarantee eventual acceptance.

We do not generally allow multiple rounds of revision so we urge you to make every effort to fully address all of the comments at this stage. If deemed necessary by the Editors, your

manuscript will be sent back to one or more of the original reviewers for assessment. If the original reviewers are not available, we may invite new reviewers.

Please submit your revised manuscript and required files (see below) no later than 21 days from today's (ie 05-Jul-2021) date. Note: the ScholarOne system will 'lock' if submission of the revision is attempted 21 or more days after the deadline. If you do not think you will be able to meet this deadline please contact the editorial office immediately.

on behalf of Dr Jake Socha (Associate Editor) and R. Kerry Rowe (Subject Editor)
openscience@royalsociety.org

Associate Editor Comments to Author (Dr Jake Socha):

The reviewers identify numerous issues that need to be addressed in the next revision. Please pay particular attention to reviewer 1's comments about the incongruence between the conclusions and the simplicity of the model. If the conclusions do not conform tightly with the data, the manuscript will not move forward to publication.

Reviewer comments to Author:
Reviewer: 1
Comments to the Author(s)

This work performed a scaling analysis on a model flapping wing system. It calculated the dependency of system design parameters and performance (e.g., actuation voltage, wing frequency, power, efficiency) on the wing actuator mass. To enable this analysis, the authors obtained empirical scaling rules of electric motors, quasi-steady aerodynamic model, and other empirical scaling rules on the wing length, propulsion-system mass, as well as using different profiles of angle of attack trajectory.

I think this is an interesting subject, the paper is clearly written and easily to follow. However, to enhance the contribution and significance of the work, I suggest the following.

1. The model used is highly, highly simplified! We know that wing design/structural dynamics have significant effects on the AoA (twisting/bending), the lift generation, efficiency etc. And its scaling is highly complex and different in animals (different species of insects, hummingbirds, etc) and potentially in robotic systems as well, and we don't know if the rigid wing assumption applies well to all scales. Also the quasi-steady model also depend on the scales (e.g., Reynolds number, which has been modeled in the literature) in addition to aspect ratio. The wing actuation mechanisms also varies from direct-motor drive with gears, to those with four-bar, to

piezoelectric and electromagnetic. The current model based only on the direct motor drive model. Considering all these complexities, I'm really skeptical that the conclusions derived based on this model could have strong practical significance. Therefore, to enhance this aspect, I would suggest 1) authors to collect the actual data on the existing designs of flapping wing robots and integrate them with the predicted performance of the system, 2) based on the above comparison/analyses, provide some guidelines or at least some suggestions for future designs.

2. I would also suggest compare with rotary wing system, to my understanding, the rotary wing system should be more efficiency in hovering flight than the flapping wing system, at least in medium to large scales. This is important to understand at which scale the flapping wing flight becomes more advantageous.

3. Please show the source of data in Fig. 2 and 4.

Reviewer: 2

Comments to the Author(s)

Please refer to the attached document (RSOS-210542 Review.pdf)

Reviewer: 3

Comments to the Author(s)

Thanks for the effort done in this manuscript. (RSOS-210452_review.pdf)

===PREPARING YOUR MANUSCRIPT===

If you have been asked to revise the written English in your submission as a condition of publication, you must do so, and you are expected to provide evidence that you have received language editing support. The journal would prefer that you use a professional language editing service and provide a certificate of editing, but a signed letter from a colleague who is a native speaker of English is acceptable. Note the journal has arranged a number of discounts for authors

using professional language editing services
(<https://royalsociety.org/journals/authors/benefits/language-editing/>).

===PREPARING YOUR REVISION IN SCHOLARONE===

<https://royalsociety.org/journals/authors/author-guidelines/#supplementary-material> to include a suitable title and informative caption. An example of appropriate titling and captioning may be found at https://figshare.com/articles/Table_S2_from_Is_there_a_trade-

off_between_peak_performance_and_performance_breadth_across_temperatures_for_aerobic_sc
ope_in_teleost_fishes_/3843624.

Author's Response to Decision Letter for (RSOS-210452.R0)

See Appendix C.

RSOS-210452.R1 (Revision)

Review form: Reviewer 1

Is the manuscript scientifically sound in its present form?

Yes

Are the interpretations and conclusions justified by the results?

Yes

Is the language acceptable?

Yes

Do you have any ethical concerns with this paper?

No

Have you any concerns about statistical analyses in this paper?

No

Recommendation?

Accept as is

Comments to the Author(s)

I appreciate the authors' efforts in addressing my previous comments, I'm happy with the revisions and to recommend acceptance.

Review form: Reviewer 2 (Moatasem Fouda)

Is the manuscript scientifically sound in its present form?

Yes

Are the interpretations and conclusions justified by the results?

Yes

Is the language acceptable?

Yes

Do you have any ethical concerns with this paper?

No

Have you any concerns about statistical analyses in this paper?

No

Recommendation?

Accept as is

Comments to the Author(s)

Thank you for your detailed replies and for considering most of my comments. I could not agree with you regarding the efficiency of the 90 deg flapping amplitude, noting it creates a clap and fling like the X-configuration at the top and lower positions. I could not agree 100% percent with your opinion. This minor disagreement would not affect the reliability of your nice work. I would like to thank you for breaking the gap between academic research and the industry by providing design schemes like this paper, which will shorten the design time of many engineers. Finally, I congratulate you on this nice and important work and I highly encourage you to resume your work in another paper to cover the performance testing of these micro air vehicles, which is another significant design phase like the preliminary design phase.

Review form: Reviewer 3 (Mohamed Zakaria)

Is the manuscript scientifically sound in its present form?

Yes

Are the interpretations and conclusions justified by the results?

Yes

Is the language acceptable?

Yes

Do you have any ethical concerns with this paper?

No

Have you any concerns about statistical analyses in this paper?

No

Recommendation?

Accept as is

Comments to the Author(s)

Thanks for addressing my comments

Decision letter (RSOS-210452.R1)

Dear Dr Nabawy,

It is a pleasure to accept your manuscript entitled "Scalability of Resonant Motor-Driven Flapping Wing Propulsion Systems" in its current form for publication in Royal Society Open Science. The comments of the reviewer(s) who reviewed your manuscript are included at the foot of this letter.

on behalf of Dr Jake Socha (Associate Editor) and R. Kerry Rowe (Subject Editor)
openscience@royalsociety.org

Associate Editor Comments to Author (Dr Jake Socha):
The authors have done a commendable job on addressing the reviewers' concerns.
Congratulations on acceptance of this article!

Reviewer comments to Author:

Reviewer: 1

Comments to the Author(s)

I appreciate the authors' efforts in addressing my previous comments, I'm happy with the revisions and to recommend acceptance.

Reviewer: 2

Comments to the Author(s)

Thank you for your detailed replies and for considering most of my comments. I could not agree with you regarding the efficiency of the 90 deg flapping amplitude, noting it creates a clap and fling like the X-configuration at the top and lower positions. I could not agree 100% percent with your opinion. This minor disagreement would not affect the reliability of your nice work. I would like to thank you for breaking the gap between academic research and the industry by providing design schemes like this paper, which will shorten the design time of many engineers. Finally, I congratulate you on this nice and important work and I highly encourage you to resume your work in another paper to cover the performance testing of these micro air vehicles, which is another significant design phase like the preliminary design phase.

Reviewer: 3

Comments to the Author(s)

Thanks for addressing my comments

Appendix A

Paper Title: Scalability of Resonant Motor-Driven Flapping Wing Propulsion Systems

Manuscript ID: RSOS-210452

Journal: Royal Society Open Science

Authors: Mostafa R. A. Nabawy and Ruta Marcinkeviciute

Review Summary:

The paper presents a design and optimization scheme for the resonant flapping wings to enable the designers of the flapping MAVs and drones to scale their designs based on some design inputs such as the wing planform and mass, the weight, the resonating spring stiffness, and the motor characteristics. The optimization technique is based on building two models: The aerodynamic model and the dynamics model. The aerodynamic model will be fed with the weight and wing planform, assuming constant flapping angle and lift to weight ratio. The outputs of this model are the optimal beating frequency, the air damping, and the aerodynamic propulsion power. The optimal beating frequency and the air damping, in addition to other design inputs such as the inertia of the motor, the transmission and the wing, the damping of the motor, and the stiffness of the resonating spring, all will be fed into the dynamics model which will calculate the required voltage and electrical power to drive this flapping propulsion wing. Finally, system efficiency will be estimated using the two outputs of the aerodynamics and dynamics model: aerodynamic power and electrical power.

The assumption of a fixed flapping amplitude φ_{max} as an input of the aerodynamics model is a major assumption, which needs more assessment. This assumption is valid for the non-oscillating motors and linkage-driven flapping mechanisms, but it is not valid for the resonant motor-driven flapping mechanisms, as explained below in Major comments 1&2.

Comments on the manuscript:

Major:

Major Comment#1 Page 4 of 19 3.1 Vehicle Configuration 2nd Paragraph and some other locations:

This is just a conflicting statement that needs to be rephrased. *“Note that, the transmission system here is a gear box that reduces the motor rotational frequency to the required flapping frequency.”*

According to the mechanism described in the 1st paragraph, this oscillatory flapping motion is caused by flipping the direction of the motor’s rotational motion through switching the DC polarity, creating an “AC”. Gearboxes reduce/increase the speed and its integral; angular displacement, but it will immediately transfer the change in the direction, assuming no

backlash. So, the gearbox will not reduce/convert the rotational frequency to the required flapping.

This takes us to the major concern in this paper. That the flapping amplitude and frequency are affected by the angular oscillation amplitude and frequency of the oscillating motor and for sure affected by other system characteristics like stiffness and damping. The angular amplitude of the motor is the integral of the rotational speed of the motor during one stroke. This speed is regulated by the voltage, the amplitude of the AC in this case. The frequency of the oscillatory motion is regulated by the rate of changing the polarity of the DC using the MOSFET, i.e., changing the frequency of the AC.

It is to be kindly noted that the hinge stoppers will limit the maximum flapping amplitude φ_{max} , but it will not determine it.

Other resonant flapping mechanisms' researchers consider variable flapping amplitude, as suggested. Please check Fig. 7 C in the following paper

<https://ieeexplore.ieee.org/stamp/stamp.jsp?tp=&arnumber=7828059>

Major Comment#2 Page 7 of 19 3.2.3 Wing Aerodynamics 3rd Paragraph:

“As such, in this work a ϕ_{max} value of 90 degrees will be adopted to ensure maximum aerodynamic efficiency.”

Wing efficiency Vs Flapping angle has a maximum to optimize. Authors should explain more how a 90 deg flapping angle will provide maximum efficiency.

Please refer to Figure 14.16 in the following paper (and other references):

<https://www.researchgate.net/publication/41057315> The Scalable Design of Flapping Micro-Air Vehicles Inspired by Insect Flight

Minor Comments

Minor Comment#1 Page 4 of 19 3.1 Vehicle Configuration 1st paragraph:

“The power system also includes power electronics needed to regulate the power flow from the energy source to the actuators”

This statement is generic and needs to be more detailed. Power flow of an “AC” means the amplitude and the frequency like any harmonic wave, while Figure 9, the summary diagram, states that the output of this optimizer is the operating voltage only. Authors should describe the electronics and how it works as an essential part in this machine.

Minor Comment#2 Page 4 of 19 3.1 Vehicle Configuration 2nd paragraph:

“commercial off-the-shelf components”.

Data about these components to be kindly stated such as model numbers and power, torque and efficiency curves, manufacturers, wing materials, etc.

Minor Comment#3 Page 6 of 19 3.2.2 Wing Kinematics 2nd paragraph:

The assumption of a sinusoidal flapping angle should be carefully considered in resonant systems with oscillatory motors like the system used in this paper. When using powerful motors and low stiffness torsional springs at low flapping frequencies, large flapping angles, and large gear ratios, the motor can accelerate quickly to its maximum rotational speed before reaching the mid-point of the flapping stroke. This will create an angular velocity and geometric AOA plateaus at the mid of the flapping stroke.

For the linkage flapping systems like those using crank-rocker mechanism and non-oscillating DC motor, the flapping angle could be considered sinusoidal for optimized linkages (never a perfect sine), regardless of the flapping frequency and amplitude, unlike these resonant mechanisms.

Minor Comment#4 Page 9 of 19 3.4 Actuator Considerations:

The actuator-related quantities in this section show that the definition of the word "Actuator" in this paper is limited to the oscillating DC motor, while in Figure 9, the spring stiffness is stated under the actuator parameters. It would be more convenient if the authors define the battery limits of the word (actuator) at the beginning. During reading the paper and before reaching this section I was having this question:

Does the word "actuator" include the torsional spring in addition to the DC motor or just the DC motor? The author may add a bracket to figure 8 as follows:

Concern about the design philosophy Page 10 of 19 4.1 Solution procedure:

“The former model uses the lift coefficient equation together with the wing planform geometric and kinematic characteristics to evaluate the required frequency that would allow a lift-to-weight ratio of unity. “

The designers may need to optimize the flapping frequency and amplitude to achieve higher or lower lift to weight ratio for certain flight maneuvers. So, the optimal control parameters (AC amplitude and frequency not only the voltage) should be figured against the required Lift to Weight ratio. This curve should be programmed on the control circuit to enable linear control inputs. Where the operator will have one throttle to control the magnitude of the lift force, while the control system will select the optimal AC frequency and amplitude to get the corresponding optimal flapping frequency and amplitude to achieve the desired lift force.

The authors' opinion about this topic is appreciated.

Appendix B

Review of “Scalability of Resonant Motor-Driven Flapping Wing Propulsion Systems”

In this paper entitled “**Simulation of flapping wings subjected to gusty inflow**”, the authors performed a direct design process to exploit the interaction between electromechanical and aerodynamic forces in a micro-wing flapping design. The aerodynamic model is based on a quasi-steady behavior, ignoring the dominant unsteady effects. System mechanics is modelled as a damped second order dynamic system operating at resonance with nonlinear aerodynamic damping within the lumped element model. DC motors are modelled using standard constitutive equations that relate motor operational parameters and voltage input. Design scaling laws are developed using available data in the literature.

Recommendation: This reviewer *does not* recommend publishing this paper in its current format. The work done is a pristine work and will add to the micro aerial vehicle design community. However, some references should be added that will strengthen author’s findings. Furthermore, the uncertainty analysis should be calculated for the first type uncertainty at least. Concerning the author’s last paragraph, it is important to address error bars repeatability and hysteresis based on the used devices and sensors, e.g. DC motor, voltage sensor, etc.. .

Minor Concerns:

1. It will be very useful for the reader to put the vehicle configuration summary in a table.
2. There is a tremendous work in the literature that has been done in this area. Some should be included to enrich the introduction section as well as discussion such as:
 - “Design Overview of a Resonant Wing Actuation Mechanism for Application in Flapping Wing MAVs”, Bolsman, C & Goosen, J.F.L. & Keulen, Fred. (2009). International Journal of Micro Air Vehicles. 1. 10.1260/175682909790291500.
 - "*Design optimization of flapping ornithopters: the pterosaur replica in forward flight*", MY Zakaria, HE Taha, MR Hajj, *Journal of Aircraft*, 53 (1), 48-59
 - "*Lift and drag of flapping membrane wings at high angles of attack*", MY Zakaria, DW Allen, CA Woolsey, MR Hajj, *34th AIAA Applied Aerodynamics Conference*, 3554

Appendix C

Responses to Reviewers' Comments

Article: "Scalability of Resonant Motor-Driven Flapping Wing Propulsion Systems"

Article reference: RSOS-210452

We thank the Editor and all Reviewers for their careful review of our paper. We appreciate the valuable comments made. All comments are addressed below.

In this document:

Reviewers' comments are reproduced in *italic* font and blue colour

Authors' responses are in normal font and black colour

Original sentences in the manuscript are reproduced in normal font and grey colour

New sentences added to the revised manuscript are in normal font and in red colour

Reviewer #1

This work performed a scaling analysis on a model flapping wing system. It calculated the dependency of system design parameters and performance (e.g., actuation voltage, wing frequency, power, efficiency) on the wing actuator mass. To enable this analysis, the authors obtained empirical scaling rules of electric motors, quasi-steady aerodynamic model, and other empirical scaling rules on the wing length, propulsion-system mass, as well as using different profiles of angle of attack trajectory. I think this is an interesting subject, the paper is clearly written and easily to follow. However, to enhance the contribution and significance of the work, I suggest the following.

We appreciate that the Reviewer has found our work interesting. The Reviewer's suggestions are all addressed below.

1. The model used is highly, highly simplified!

We agree that the design framework presented in our paper is simple; however, we believe this is the *adequate* level for such stage of a design process. The paper has stressed in several locations that it aims to provide a *conceptual* design process/framework for the class of vehicles considered. Conceptual design is, by definition, a low fidelity type of study, and the level of detail in modelling and configuration design is not very deep at such design stage.

We know that wing design/structural dynamics have significant effects on the AoA (twisting/bending), the lift generation, efficiency etc. And its scaling is highly complex and different in animals (different species of insects, hummingbirds, etc) and potentially in robotic systems as well, and we don't know if the rigid wing assumption applies well to all scales.

We agree with the Reviewer that structural dynamics of a wing can have significant effects on its bending/twisting behaviour and hence affect angle of attack distribution, lift generation, and efficiency. However, this is more pronounced in biological wings such as those of insects. For the robotic systems solely considered in this study, it is

a typical practice to manufacture their wings from a high stiffness carbon frame covered by a high strength thin membrane of mylar or similar. Wings constructed in this way are usually reported to have excellent strength to weight ratio. For example, Zhang and Deng reported on their Purdue resonant motor-driven design (see Ref [15] of the paper) that “The wing frame is extremely strong”. This is just an example, and all research groups building robotic insects (at all scales: from the smallest such as the Harvard Microrobotic Fly to the largest such as the Purdue design) almost adopt the same approach in building wings and all have reported high strength characteristics. As such, for the robotic insect cases which are the concern of this study, the assumption of rigid wings is acceptable.

To recognise the Reviewer’s point in our study we have now added the following sentences to the revised manuscript in page 5 as follows:

It is understood that the structural dynamics of biological wings, such as those of real insects, can significantly affect the wing bending and twisting and hence the angle of attack distribution and its corresponding aerodynamic characteristics. On the other hand, wings of robotic insects (realised at different scales) are typically made of a high strength membrane film spanning a stiffening structure made of carbon rods. These wings are always reported to have excellent strength-to-weight ratios and hence the assumption that the wings are rigid is deemed acceptable.

Also the quasi-steady model also depend on the scales (e.g., Reynolds number, which has been modeled in the literature) in addition to aspect ratio.

Reynolds number: We agree with the Reviewer that significant changes to Reynolds number will affect aerodynamic modelling accuracy. However, for the Reynolds number range considered in the current study, it is well-confirmed in the literature that the Reynolds number effect is minor. We have added the following to clarify this point in page 7 of the revised manuscript:

It is worth mentioning that the Reynolds number, Re , will have minor/negligible influence on the used aerodynamic coefficient relations within the range $1000 \leq Re \leq 15000$ [36]. This is the range that the scaling assessment presented in this study acts within. At Reynolds numbers higher than this range, flows usually get dominated by turbulent mixing, destabilising the leading-edge vortex and limiting the maximum attainable lift coefficient values [37,38]. On the other hand, at Reynolds numbers lower than this range, viscous effects become dominant and their influence cannot be neglected [30,39].

Aspect ratio: We assure the reviewer that the aspect ratio effect is already included in the aerodynamic relations used to evaluate the aerodynamic coefficients in this work. This can be seen through Equation 11 of the manuscript, and this represents an improvement on the practice usually adopted by various groups in previous robotic insect modelling work, as it is a common practice to adopt the experimental results from Dickinson for the fruit fly case and use them as a single representation for wing aerodynamic coefficients irrespective to variations in wing shape.

The wing actuation mechanisms also varies from direct-motor drive with gears, to those with four-bar, to piezoelectric and electromagnetic. The current model based only on the direct motor drive model.

We accept that our conceptual design framework is only for direct motor drive vehicles. This should not be seen as a weakness; to the opposite, it shows the adequacy of our work. Consider, for example, the conceptual design processes of jet powered and propeller driven aircraft. Whilst there can be some common elements in their established design tools, the conceptual design processes of these cases vastly vary due to the difference in the propulsion system adopted. Similarly, here, different actuation options (piezoelectrics, electromagnetics, DC motors, soft muscles, etc.) each must have an independent design process that appropriately account for its unique actuator dynamics/physics. It is inadequate, and actually impossible, to have one conceptual design process that fits all actuation options. Our work provides such a conceptual design framework for resonant, direct motor-driven vehicles which is a class of vehicles that has recently shown successful lift-off in a number of previous contributions by research groups and hence deserves to have a conceptual design framework to aid future designers to achieve speedy and better design decisions at the start of their vehicle's development.

Considering all these complexities, I'm really skeptical that the conclusions derived based on this model could have strong practical significance.

We hope we have done a good job in our above responses to eliminate the concerns the Reviewer had for the above points. Whilst we appreciate ALL of the Reviewer's concerns and we believe that the Reviewer is correct to raise them, we confirm that ALL of these concerns were seriously considered by us before developing this work: Whilst each concern usually leads to an assumption within our framework, we believe as explained above that these adopted assumptions are acceptable and do not significantly affect the adequacy of our adopted approach.

Therefore, to enhance this aspect, I would suggest 1) authors to collect the actual data on the existing designs of flapping wing robots and integrate them with the predicted performance of the system,

Despite that our work is proposing a conceptual design process rather than a modelling tool, we agree with the Reviewer that using some actual data from existing designs and comparing them to our predictions can be useful. However, it is important to draw attention to the fact that there are currently two designs of resonant motor-driven flapping wing propulsion systems available, that have relatively sufficient information. These two designs are: (1) the Carnegie Mellon University Design (based on Reference: Hines L, Campolo D, Sitti M. 2014. Liftoff of a motor-driven, flapping-wing microaerial vehicle capable of resonance. *IEEE Trans. Robot.* **30**, 220–232); and (2) the Purdue University Design (based on Reference: Zhang J, Deng X. 2017. Resonance principle for the design of flapping wing micro air vehicles. *IEEE Trans. Robot.* **33**, 183–197). We have now added the following comparison paragraph at the end of section 3.5 (page 9) of the revised manuscript as follows:

It is useful to assess how the proposed conceptual design framework predictions compare to some of the available data of current existing designs. Here, we consider

the two designs that rely on resonance and directly drive the wings through DC motor actuation, i.e. the designs from Carnegie Mellon [13] and Purdue [15] Universities. In this assessment, we will input to our framework the main inputs of these designs and evaluate the power that the motor is expected to deliver, P_{req} . There are two main reasons for picking P_{req} for assessment. First, in the absence of any efficiency figures for the two designs under consideration, P_{req} represents the final output; hence, providing judgement on the efficacy of the whole process. Second, and more importantly, the value of the power that the motor is expected to deliver, P_{req} , is arguably the most important piece of information that a designer would wish to know at the start of the design process as it enables the selection of suitable motor candidates for the vehicle under development.

For the Carnegie Mellon design, the following values are used [13]: actuator mass, m_a , of 1g; total lift measured which is used to estimate the propulsion system mass, m_p , of 3.16g (lift-to-weight ratio of 1.17); wing aspect ratio, AR , of ~3.5; maximum flapping angle amplitude, ϕ_{max} , of 70 degrees; and angle of attack value at mid half-stroke, $\alpha_{g,mid}$, of 45 degrees. A P_{req} value of 0.427W is predicted for the constant angle of attack waveform, comparing reasonably well with the reported measured value of 0.4W [13]. As for the Purdue design, we used the reported values in [15]: m_a of 2.5g; an equivalent m_p of 6.7g; AR of 4.18; ϕ_{max} of 55 degrees; and $\alpha_{g,mid}$ of 45 degrees. A P_{req} value of 1.23W is predicted for the constant angle of attack waveform, which is in good agreement with the reported measured value of ~1.2W [15]. These comparisons show that the proposed design tool can reasonably inform the designer of the power the motors should deliver, enabling a guided selection from the possible motor models at the relevant scale.

2) based on the above comparison/analyses, provide some guidelines or at least some suggestions for future designs.

Following the Reviewer's suggestion, we have provided guidelines for designing the class of vehicles in hand. This is added as a new paragraph to the revised paper's discussion in pages 14-15, as follows:

Following from the above discussions, it is informative to provide some design guidelines and suggestions for future designs:

- (1) The design of an air vehicle would normally start by considering the basic geometric characteristics of its wings. For resonant flapping wing vehicles, the developed scaling law in Eqn. 1 can provide a reasonable starting point to determine the wing scale. On the other hand, the selection of the wing aspect ratio is considered vital as it defines the severity of transformation from 2D to 3D wing aerodynamics. However, choices here are limited and only a narrow range of 3-4 is deemed practical. A wing with aspect ratio higher than four mainly suffers from stall particularly at wing regions close to the tip, whereas wings with aspect ratios lower than three will experience reduced aerodynamic performance due to increased 3D induced aerodynamic effects. Wing chord distribution can be considered at a later stage of the design process to tune the aerodynamics to desired performances (e.g. as proposed in [20]) depending on vehicle requirements.

- (2) Following wing geometric considerations, wing kinematics should be decided. A sinusoidal variation of the wing flapping motion is a practical choice for resonant flapping wing vehicles; however, the amplitude of the flapping angle depends on the actuator involved and the degree of sophistication of its associated transmission mechanism. For example, piezoelectric or electromagnetic actuated designs typically involve relatively sophisticated transmission mechanisms unable to allow the reach to a maximum flapping amplitude angle of 90 degrees. However, resonant motor-driven vehicles, considered in this study, can practically achieve such a maximum amplitude value. This, in turn, allows the wing to scan a larger flapping disk area and hence improve the aerodynamic efficiency of lift production.
- (3) Pitching kinematics are typically realized with passive wing hinges, and designers can opt to design these hinges either to enable higher lift coefficient values if a constant angle of attack of 45 degrees can be maintained through the half-strokes or to allow higher system efficiency values if the hinge is designed to allow a sinusoidal variation of the angle of attack.
- (4) The system elastic element should be selected, for a given system inertia, to allow system resonance at a frequency that would enable sufficient wing speed to satisfy the lift production demand.
- (5) The gearbox should be selected to allow appropriate gear ratio between the wing speed and the recommended speed for motor operation in such a way that the latter speed is not exceeded under different operating conditions.
- (6) Motor selection is primarily decided by the ability of a motor candidate to deliver the required power for the application, and current motors with masses up to 1-2 gm seem to be able to deliver sufficient power for the resonant motor-driven flapping wing vehicle application.
- (7) Finally, a resonant motor-driven flapping wing vehicle is a more efficient system at small scales; however, as scale increases, switching to a rotary wing system becomes significantly more efficient.

2. I would also suggest compare with rotary wing system, to my understanding, the rotary wing system should be more efficiency in hovering flight than the flapping wing system, at least in medium to large scales. This is important to understand at which scale the flapping wing flight becomes more advantageous.

Following the Reviewer's suggestion, we have added two new parts to the revised manuscript. The first part is to explain the model for a rotary wing system, whereas the second part is within the results section to discuss the results of the comparison. These two parts are reproduced below.

Modelling part (page 10 of the revised manuscript):

3.6 Rotary wing system

It is instructive to compare the efficiency levels from the flapping system described above with an analogous rotary system. This comparison is useful in providing a *preliminary* assessment of the relative advantage of different flying modes (flapping vs rotary) at different scales. In evaluating the efficiency levels for rotary actuation, we assume that the same vehicle configuration will hold, i.e. the vehicle still has two motors and each motor is driving the exact same wing used in flapping, but now in a rotary fashion. This way, it is ensured that all other elements of the comparison are

kept the same, and only wing motion is varied. However, from a configuration point of view, such rotary system is not realised in such way as rotary vehicles typically adopt a propeller-like wing and usually adopt more than two motors. That said, this does not detract from the usefulness of this comparison as the main aim here is to understand the system efficiency levels when the same actuator operates in flapping or rotary motion under the same conditions.

For constant rotational speed and constant angle of attack, i.e. rotary motion, the aerodynamic power expended by a single wing and the frequency expressions can be evaluated based on [30]:

$$P_{aero,rot} = W^{\frac{3}{2}} \sqrt{\frac{2}{\rho S} \frac{C_D}{C_L^2} \left(\frac{\hat{r}_3}{\hat{r}_2}\right)^3} \times \frac{1}{2} \quad (30)$$

$$f_{rot} = \sqrt{\frac{2W}{\rho(2\pi R_w \hat{r}_2)^2 (2R_w^2/AR) (C_{L\alpha} \sin(\alpha_g) \cos(\alpha_g))}} \quad (31)$$

where W is the weight of the whole system, S is the area of both wings, and α_g is the geometric angle of attack set as 45 degrees for consistency. Note that, both rotary (also so-called revolving) and flapping wing motions share the same expression for the lift coefficient [30,31,34]. Direct comparison of Eqn. (30) with Eqn. (15) shows that the rotary aerodynamic power expression is 20% less in value compared to sinusoidal flapping with constant angle of attack. This implies that rotary motion will expend less aerodynamic power to support the same weight, and thus will always be aerodynamically more efficient. Once motion frequency is evaluated using Eqn. (31), values for gearhead transmission ratio, $N_{g,rot}$, and gearhead efficiency, $\eta_{g,rot}$, for the rotary system can be evaluated in a similar fashion to using Eqns 19-21.

The torque of the rotary system, T_{rot} , at the motor output point can be evaluated from:

$$T_{rot} = \frac{P_{aero,rot}}{\eta_{g,rot} \Omega_{rec}} \quad (32)$$

where Ω_{rec} is the motor recommended rotation speed as defined in Eqns 20 and 21. The motor operating current can, thus, be obtained as:

$$I_{rot} = \frac{T_{rot}}{K_a} + I_0 \quad (32)$$

where I_0 is the no-load current which for the series of relevant motors: 0206, 0308, 0615, 0816, 1016, and 1024 from Faulhaber [40], a fitting relation can be produced to represent it as:

$$I_0 [Amp] = 0.1998 m_a^{-0.359} [mg], n = 6, R^2 = 0.85, s_{I_0/m_a} = 0.0054 Amp \quad (33)$$

The motor supply voltage is, then, calculated as:

$$V_{rot} = R_a I_{rot} + K_a \Omega_{rec} \quad (34)$$

The rotary system efficiency can, therefore, be obtained as:

$$\eta = \frac{P_{aero,rot}}{V_{rot}I_{rot}} \quad (35)$$

Results part (page 13 of the revised manuscript):

Finally, Fig. 12 compares the efficiency levels from the flapping wing system considered in this work to that obtained from the conceived rotary wing system presented in Section 3.6. In this comparison, we show results for μ_a values representing the bounds of the range considered in this study. It is evident that increasing μ_a increases the system efficiency values and move the efficiency peaks towards higher actuator mass values. However, μ_a has no influence on the general trends of the system efficiency curves, hence does not influence the main conclusion from this demonstration. The main outcome of this demonstration is that at small scales, the system efficiency levels of the flapping system described in this work are slightly higher than that when a rotary system is adopted. In fact, the efficiency values of the rotary system deteriorate at small scales and this is mainly due to the higher no-load current, I_0 , values as the motor scale decreases. On the other hand, as scale increases, the rotary system efficiency levels increase, significantly surpassing the flapping system efficiency levels. This result is in line with the findings of Hawkes and Lentink [41] who showed that flapping wing vehicles actuated with piezoelectric actuators have higher system efficiency when compared to rotary motor-actuated vehicles at small scales similar to tiny insects and this trend flips as scale increases.

Figure 12: Comparison between efficiency levels of flapping and rotary wing systems. AoA denotes angle of attack waveform.

3. Please show the source of data in Fig. 2 and 4.

We have now added clarifying sentences in text to clearly show the source of the data as follows:

In Page 4: Figure 2a shows a log-log plot of the data collected for up-to-date successful designs – The data points shown in Figure 2a are provided in Table 1 in which the source of each data point is provided.

In Page 5: **Figure 2b shows the wing structural mass data sourced from [26] together with our developed scaling law**

In Page 6: **Figure 4 shows the data sourced from [22] for the non-dimensional radius for first, second, and third moment of wing area.**

Reviewer #2

Review Summary: *The paper presents a design and optimization scheme for the resonant flapping wings to enable the designers of the flapping MAVs and drones to scale their designs based on some design inputs such as the wing planform and mass, the weight, the resonating spring stiffness, and the motor characteristics. The optimization technique is based on building two models: The aerodynamic model and the dynamics model. The aerodynamic model will be fed with the weight and wing planform, assuming constant flapping angle and lift to weight ratio. The outputs of this model are the optimal beating frequency, the air damping, and the aerodynamic propulsion power. The optimal beating frequency and the air damping, in addition to other design inputs such as the inertia of the motor, the transmission and the wing, the damping of the motor, and the stiffness of the resonating spring, all will be fed into the dynamics model which will calculate the required voltage and electrical power to drive this flapping propulsion wing. Finally, system efficiency will be estimated using the two outputs of the aerodynamics and dynamics model: aerodynamic power and electrical power.*

The assumption of a fixed flapping amplitude ϕ_{max} as an input of the aerodynamics model is a major assumption, which needs more assessment. This assumption is valid for the non-oscillating motors and linkage-driven flapping mechanisms, but it is not valid for the resonant motor-driven flapping mechanisms, as explained below in Major comments 1&2.

We thank the Reviewer for their thorough review of our work. All concerns of the Reviewer are addressed below.

Major Comment#1_ Page 4 of 19_ 3.1 Vehicle Configuration_2nd Paragraph and some other locations:

This is just a conflicting statement that needs to be rephrased. “Note that, the transmission system here is a gear box that reduces the motor rotational frequency to the required flapping frequency.”

According to the mechanism described in the 1st paragraph, this oscillatory flapping motion is caused by flipping the direction of the motor’s rotational motion through switching the DC polarity, creating an “AC”. Gearboxes reduce/increase the speed and its integral; angular displacement, but it will immediately transfer the change in the direction, assuming no backlash. So, the gearbox will not reduce/convert the rotational frequency to the required flapping.

We thank the Reviewer for highlighting this issue. We appreciate that the sentence was not accurate enough and has now been improved (in page 3 of the revised manuscript) as follows:

The AC signal supplied to the motor ensures it flips its rotational direction generating the oscillatory flapping motion, and the gear box transmission system immediately transfers this change in direction whilst modulating speed.

This takes us to the major concern in this paper. That the flapping amplitude and frequency are affected by the angular oscillation amplitude and frequency of the oscillating motor and for sure affected by other system characteristics like stiffness and damping. The angular amplitude of the motor is the integral of the rotational speed of the motor during one stroke. This speed is regulated by the voltage, the amplitude of the AC in this case. The frequency of the oscillatory motion is regulated by the rate of changing the polarity of the DC using the MOSFET, i.e., changing the frequency of the AC.

It is to be kindly noted that the hinge stoppers will limit the maximum flapping amplitude ϕ_{max} , but it will not determine it.

Other resonant flapping mechanisms' researchers consider variable flapping amplitude, as suggested. Please check Fig. 7 C in the following paper <https://ieeexplore.ieee.org/stamp/stamp.jsp?tp=&arnumber=7828059>

We recognise that other researchers (such as in the reference pointed out by the Reviewer) have considered their vehicle's performance over a range of flapping frequencies and their corresponding flapping angle amplitude. However, these studies were focusing on simulating and/or testing the performance of their vehicles against different input excitations. However, our study has a completely different goal in that it addresses how to design such class of vehicles rather than to assess them against different inputs. In other words, other studies are concerned with analysing how much lift will be generated for a given input electric signal for a given design – our work does the opposite as it defines the input electric signal and system requirements to satisfy a specific lift demand at a range of scales. As such, we have decided to fix the flapping angle magnitude to a fixed value known to enhance the *aerodynamic* efficiency (please see our detailed explanation of this selection in our response to the next review point). For this given flapping amplitude value, the flapping frequency that would allow weight balance is determined. It has been proved that maximum system efficiency is achieved when the system is driven at the natural frequency; for a detailed explanation, please refer to reference “Zhang J, Deng X. 2017. Resonance principle for the design of flapping wing micro air vehicles. *IEEE Trans. Robot.* **33**, 183–197”. As such, we equate the calculated flapping frequency to the system natural frequency to obtain the stiffness of the elastic element; in other words, we obtain the stiffness of the elastic element that would allow the system natural frequency to be equal to the flapping frequency required for weight balance. All these information and others are fed into the dynamics model that eventually provides the input electric AC signal waveform that is needed to ultimately achieve the required flapping motion. Hence, the starting point of our design process is to define an “target” flapping angle amplitude and evaluate the frequency required to ensure successful operation for this defined angle amplitude. This provides a “design” point to work for, and is acceptable as a design process only needs one design point, and should not be concerned with other “off-design” points. Clearly, the design point selection can vary from a designer to a designer, and here we opted to make a selection that has its *aerodynamic* merits. On the other hand, the off-design points can be assessed in a later stage if the performance at such off-design operating conditions (other frequencies and amplitudes) is of interest.

We would like to note that hinge stoppers are to control the maximum amplitude of the angle of attack and has no influence on the flapping angle amplitude.

Major Comment#2_ Page 7 of 19_ 3.2.3 Wing Aerodynamics_3rd Paragraph:
“As such, in this work a ϕ_{max} value of 90 degrees will be adopted to ensure maximum aerodynamic efficiency.”

Wing efficiency Vs Flapping angle has a maximum to optimize. Authors should explain more how a 90 deg flapping angle will provide maximum efficiency.

Please refer to Figure 14.16 in the following paper (and other references):

<https://www.researchgate.net/publication/41057315> The Scalable Design of Flapping Micro-Air Vehicles Inspired by Insect Flight

We appreciate the reference that the Reviewer has pointed. Figure 14.16 of this suggested reference assesses the efficiency in terms of lift to power ratio. However, the power measurements used to produce this figure are for the total power including the aerodynamic power as well as the power needed to drive the motor, the drive train, and overcome the inertia of the complete mechanism. The figure is, therefore, not indicative of the aerodynamic power alone; hence, cannot be used to assess the aerodynamic efficiency. More importantly, these measurements were conducted on the DelFly, a configuration with four wings in the so-called X-wing configuration, and the DelFly aerodynamics rely to a huge extent on the clap and fling aerodynamic mechanism. As such, the aerodynamics involved are irrelevant to those that would be experienced by the configurations considered in our study.

It should be noted that we stated that the selection of the flapping amplitude is to ensure maximum *aerodynamic* efficiency (not system efficiency). Hence, the decision is solely driven by *aerodynamics* considerations. Moreover, having a higher ϕ_{max} is advantageous in that it enables primary flow features (such as the leading-edge vortex) to get established and unsteady flow features to diminish.

To clarify why a flapping angle amplitude of 90° would lead to maximum aerodynamic efficiency, one way of explaining it is to consider the well-established Rankine-Froude momentum theory and how it is applied within the context of hovering flight. Consider the figure below which demonstrates the “flapping disk area” used in simple momentum theory modelling.

The induced downwash velocity, w_i , for a hovering insect/vehicle at the actuator disk is known to be:

$$w_i = \sqrt{\frac{W}{2\rho A}}$$

where W is the total weight, ρ is the air density and A is the flapping disk area. The induced aerodynamic power, P_i , is thus obtained as:

$$P_i = \sqrt{\frac{W^3}{2\rho A}}$$

It is, thus, evident that as the flapping disk area increases both the induced downwash and consequently the induced aerodynamic power decreases, directly translating to an improved aerodynamic efficiency of lift production (i.e. producing a given amount of lift with less induced power expenditure). The disk area, A , for a flapping wing with a horizontal stroke plane is defined as (see above figure):

$$A = 2\phi_{max}R^2$$

where the factor 2 is to account for the two wings. Clearly, as ϕ_{max} approaches 90° (i.e. $\pi/2$), the flapping disk area reaches its maximum value corresponding to the circular disk area of πR^2 . As such, using a flapping angle amplitude of 90° ensures maximum disk area and hence minimum induced downwash and induced aerodynamic power to produce a given amount of lift, leading to maximum *aerodynamic* efficiency.

Additionally, as explained in the original manuscript, the value of k_{flap} which accounts for the reduction in the actuator disk area approaches unity (ideal value) as ϕ_{max} approaches 90° . Figure 5 of the manuscript showed how the lift curve slope can change against different values of ϕ_{max} due to the effect of k_{flap} , and it was shown that significant variations in the flapping angle amplitude could have a significant influence on k_{flap} and hence the aerodynamic coefficient values.

To better explain this point in the manuscript, the following clarification has been added to pages 6-7 of the revised manuscript:

Clearly, as ϕ_{max} approaches 90° , the flapping actuator disk area reaches its maximum value. Considering the Rankine-Froude momentum theory, the increase in the actuator disk area to its maximum value leads the induced downwash velocity and consequently the induced aerodynamic power expended whilst producing a given amount of lift to achieve their minimum values [19,35]. This is reflected in Eqn. (12) as k_{flap} approaches unity (ideal value) when ϕ_{max} approaches 90 degrees. Figure 5 shows how $C_{L\alpha}$ can change against different values of ϕ_{max} due to change in k_{flap} . It could be seen that a significant decrease in the flapping angle amplitude will reduce k_{flap} and hence significantly deteriorates the aerodynamic coefficient values. As such, based on the previous, in this work a ϕ_{max} value of 90 degrees will be adopted to ensure maximum *aerodynamic* efficiency.

Minor Comment#1_ Page 4 of 19_ 3.1 Vehicle Configuration_1st paragraph:

“The power system also includes power electronics needed to regulate the power flow from the energy source to the actuators”

This statement is generic and needs to be more detailed. Power flow of an “AC” means the amplitude and the frequency like any harmonic wave, while Figure 9, the summary diagram, states that the output of this optimizer is the operating voltage only. Authors should describe the electronics and how it works as an essential part in this machine.

We agree with the Reviewer that the power electronics is an essential subsystem of the fully integrated vehicle. However, the focus here is on the design of the propulsion system only, i.e. wings, transmissions, and actuators. As such, the general functionality of the power electronics is addressed here without going into the details of circuit design (which is out of the focus of the current study). We have now included a couple of sentences to page 3 of the revised manuscript to better describe the role of the power electronic circuit as follows:

The power system also includes power electronics needed to regulate the power flow from the energy source to the actuators. **Hence, the power electronics circuit does two roles: (1) it converts the DC output from the power source to an AC input to the actuators ensuring that the required AC waveform variation and driving frequency are produced; and (2) it modulates the amplitude of the AC voltage to ensure successful operation.**

It is to be noted that the design process presented in this work, and summarised in Figure 9, has a number of outputs. The operating voltage is one of them and represents the amplitude of the AC waveform to be inputted to the motor. The process also outputs the operational frequency (i.e. system resonant frequency); hence, both the amplitude and frequency of the input signal are determined/evaluated from the process.

We recognise describing the voltage in Figure 9 by the word “operating” can be confusing and can be misunderstood as the DC voltage. To avoid such confusion, the voltage is now described in Figure 9 as “AC Voltage Amplitude”.

Minor Comment#2_ Page 4 of 19_ 3.1 Vehicle Configuration_2nd paragraph:

“commercial off-the-shelf components”.

Data about these components to be kindly stated such as model numbers and power, torque and efficiency curves, manufacturers, wing materials, etc.

We have added more information of the example vehicle configuration. However, we would like to stress that this paper is concerned with developing a conceptual design framework for resonant motor-driven flapping wing propulsion systems and is not specific for a certain vehicle. This is why we deliberately did not want to include lots of details (such as model numbers and performance curves) of the example vehicle shown in the paper to avoid deviating readers’ attention from the main aim of the work. However, following the Reviewer’s recommendation, more information on the vehicle configuration has now been added to page 3 of the revised manuscript as follows:

A video is provided in the supplementary material for the propulsion system shown in Figure 1 built using commercial off-the-shelf components. **This propulsion system**

weighs 3.4g. Each motor, weighing 1g, had its gear box, weighing 0.2g, fully integrated to it. An elastic element in the form of an elastic band was used. The wings had an elliptic planform shape and were realised using a polyimide film spanning on top of a rigid structure of carbon fibre rods. The passive hinge required to ensure adequate wing pitching was realised using a simple way of adding an extra carbon fibre rod parallel to the leading-edge rod. As such, the membrane material enclosed between the two rods provided the hinge functionality. This method is easy in prototyping; however, the extra carbon fibre rod adds weight to the wing structure, which is less favourable for flapping wings. Wing stoppers were added to avoid angles of attack more than 45 degrees. This propulsion system was able to demonstrate tethered operation; however, it is important to stress that this propulsion system is included here just to demonstrate the configuration and functionality of such type of concepts. In fact, this paper will focus on developing a generic conceptual design tool for resonant motor-driven flapping propulsion system and is not specific for a certain configuration.

Minor Comment#3_ Page 6 of 19_ 3.2.2 Wing Kinematics_2nd paragraph:

The assumption of a sinusoidal flapping angle should be carefully considered in resonant systems with oscillatory motors like the system used in this paper. When using powerful motors and low stiffness torsional springs at low flapping frequencies, large flapping angles, and large gear ratios, the motor can accelerate quickly to its maximum rotational speed before reaching the mid-point of the flapping stroke. This will create an angular velocity and geometric AOA plateaus at the mid of the flapping stroke.

For the linkage flapping systems like those using crank-rocker mechanism and non-oscillating DC motor, the flapping angle could be considered sinusoidal for optimized linkages (never a perfect sine), regardless of the flapping frequency and amplitude, unlike these resonant mechanisms.

We appreciate the Reviewer's concern; however, we assure the Reviewer that the sinusoidal flapping waveform has always been adopted and proved to be adequate for resonant systems with oscillatory motors (such as the vehicles of concern in this study). This has been shown by different groups that have tested such type of systems. To provide a couple of examples, the Reviewer is referred to Figure 4 of reference: "Campolo D. 2010. Motor selection via impedance-matching for driving nonlinearly damped, resonant loads. *Mechatronics* **20**, 566–573". In that figure it was shown that the relative errors between sinusoidal and actual values is within 3% for flapping angle and 5% for angular velocity, justifying the sinusoidal assumption.

Another example is the work done by the Purdue group who experimentally demonstrated the flapping angle variation of their resonant oscillating motor system in Figure 7a of reference: "Zhang J, Deng X. 2017. Resonance principle for the design of flapping wing micro air vehicles. *IEEE Trans. Robot.* **33**, 183–197." In that figure, the Purdue group showed that the experimentally measured flapping angle has a clear sinusoidal variation, justifying the sinusoidal flapping angle variation assumption.

As a side note, from an aerodynamic point of view, it is known that the lift force is related to the square of the flapping angular velocity, $(\dot{\phi})^2$. Assuming a constant angle of attack variation for simplicity, the average lift over a flapping cycle will be proportional to the mean square of the flapping angular velocity over the flapping cycle.

As such, for a given flapping amplitude and frequency, the sinusoidal waveform of the flapping angle will lead to a higher mean square velocity value, hence would allow higher lift forces to be generated, which is much needed to achieve successful lift-off.

We find it useful to briefly highlight the above point in the manuscript; as such, the following has been added to the revised manuscript in page 5:

The flapping angle time variation, $\phi(t)$, is considered based on a sinusoidal waveform which is a realistic representation of typical flapping variations within real and/or robotic insects. In fact, previous investigations of resonant motor-driven flapping vehicles have clearly demonstrated the accuracy of the adopted sinusoidal variation in resembling actual measured flapping angle variations of these systems [15,17].

Minor Comment#4 Page 9 of 19 3.4 Actuator Considerations:

The actuator-related quantities in this section show that the definition of the word "Actuator" in this paper is limited to the oscillating DC motor, while in Figure 9, the spring stiffness is stated under the actuator parameters. It would be more convenient if the authors define the battery limits of the word (actuator) at the beginning. During reading the paper and before reaching this section I was having this question: Does the word "actuator" include the torsional spring in addition to the DC motor or just the DC motor? The author may add a bracket to figure 8 as follows:

We thank the Reviewer for highlighting this issue. We agree with the Reviewer that the word "Actuator" was not clearly explained and the naming of section headers in section 3.4 can be confusing. To clarify, here, we use the word "Actuator" to refer to the "DC Motor" only. To better clarify this point, we have now added the following sentence to section 3.4 (page 8 of the revised manuscript) as follows:

Note that, here, actuator refers to the DC motor.

For even better clarity, we have also separated Section 3.4 into two sections: Section 3.4 dealing with actuator scaling laws and 3.5 dealing with system dynamics and efficiency.

We assure the Reviewer that in Figure 9 the stiffness block has a green colour which designates system parameters, as opposed to yellow colour that designates actuator parameters, hence, we believe Figure 9 is correct.

We agree with the Reviewer that Figure 8's clarity can be improved, and we have now added the word "Actuator" beside "Motor", as recommended by the reviewer as follows:

Figure 8: Schematic diagram for the modelling representation of the main system components. For compactness, only a part of the wing is shown.

Concern about the design philosophy _ Page 10 of 19_ 4.1 Solution procedure:

“The former model uses the lift coefficient equation together with the wing planform geometric and kinematic characteristics to evaluate the required frequency that would allow a lift-to weight ratio of unity. “

The designers may need to optimize the flapping frequency and amplitude to achieve higher or lower lift to weight ratio for certain flight maneuvers. So, the optimal control parameters (AC amplitude and frequency not only the voltage) should be figured against the required Lift to Weight ratio. This curve should be programmed on the control circuit to enable linear control inputs. Where the operator will have one throttle to control the magnitude of the lift force, while the control system will select the optimal AC frequency and amplitude to get the corresponding optimal flapping frequency and amplitude to achieve the desired lift force.

The authors’ opinion about this topic is appreciated.

The Reviewer is correct that both flapping frequency and amplitude can be changed simultaneously to modulate (increase or decrease) the amount lift to weight ratio, and that this is essentially needed for manoeuvring flight. However, manoeuvres are considered “off-design” points that can be considered at a later stage of a vehicle’s development. This work is concerned with *conceptual* design and hence only deals with the “design” point of the vehicle, and the main aim of a design point is to achieve successful lift-off in the first place. Once a vehicle has been able to demonstrate successful lift-off in hover, off-design considerations including manoeuvres should be assessed. Given the previous, to define our design point, we opted to define a maximum flapping angle that would ensure maximum aerodynamic efficiency (as explained in our above response) and evaluate the required flapping frequency to ensure successful lift-off. Clearly, depending on design constraints, a designer may decide a different approach (defining frequency and calculating amplitude or calculating both in a simultaneous fashion to achieve a certain objective). However, in the absence of particular constraints and to achieve highest level of generality, we believe that our proposed approach is probably the most sensible. We believe that the

Reviewer's point is important and needs clarification for readers, hence we have included clarification sentences to the revised manuscript in page 11 as follows:

The former model uses the lift coefficient equation together with the wing planform geometric and kinematic characteristics to evaluate the required frequency that would allow a lift-to-weight ratio of unity. The aerodynamic model, also, evaluates the aerodynamic drag/damping, B_{s2} , required within the system dynamics model as well as the aerodynamic power needed to evaluate the system efficiency. **It should be noted that the current aerodynamic model considers lift-off as its primary objective. Hence, the flapping angle amplitude is fixed to its maximum possible value to ensure highest aerodynamic efficiency, whereas only flapping frequency is evaluated to guarantee sufficient lift is generated to balance weight. In the absence of any known design constraints, this approach is deemed adequate (or at least sufficient); however, designers may opt to evaluate the flapping amplitude for a constrained frequency or evaluate both frequency and flapping amplitude to achieve a certain design objective. Moreover, the combined modulation of both flapping frequency and amplitude should be identified for "off-design" conditions, e.g. in maneuvering flight, to control the amount of aerodynamic force generated. Ultimately, optimum combinations of flapping frequency and amplitude for different flight conditions and their corresponding input electric signal parameters should be defined. These can then be programmed within the controller circuit to ensure that optimum parameters are always employed for the different flight scenarios.**

Reviewer #3

In this paper entitled "Simulation of flapping wings subjected to gusty inflow", the authors performed a direct design process to exploit the interaction between electromechanical and aerodynamic forces in a micro-wing flapping design. The aerodynamic model is based on a quasi-steady behavior, ignoring the dominant unsteady effects. System mechanics is modelled as a damped second order dynamic system operating at resonance with nonlinear aerodynamic damping within the lumped element model. DC motors are modelled using standard constitutive equations that relate motor operational parameters and voltage input. Design scaling laws are developed using available data in the literature.

Recommendation: *This reviewer **does not** recommend publishing this paper in its current format. The work done is a pristine work and will add to the micro aerial vehicle design community. However, some references should be added that will strengthen author's findings. Furthermore, the uncertainty analysis should be calculated for the first type uncertainty at least. Concerning the author's last paragraph, it is important to address error bars repeatability and hysteresis based on the used devices and sensors, e.g. DC motor, voltage sensor, etc.. .*

Minor Concerns:

- 1. It will be very useful for the reader to put the vehicle configuration summary in a table.*
- 2. There is a tremendous work in the literature that has been done in this area. Some should be included to enrich the introduction section as well as discussion such as:
- "Design Overview of a Resonant Wing Actuation Mechanism for Application in Flapping Wing MAVs", Bolsman, C & Goosen, J.F.L. & Keulen, Fred. (2009). International Journal of Micro Air Vehicles. 1. 10.1260/175682909790291500.*

- "Design optimization of flapping ornithopters: the pterosaur replica in forward flight", MY Zakaria, HE Taha, MR Hajj, *Journal of Aircraft*, 53 (1), 48-59
- "Lift and drag of flapping membrane wings at high angles of attack", MY Zakaria, DW Allen, CA Woolsey, MR Hajj, 34th AIAA Applied Aerodynamics Conference, 3554

We appreciate that the Reviewer has found our work novel and adding value to the micro air vehicle design community.

Following the Reviewer's suggestions, all references proposed by the Reviewer are now included to the discussion on page 13 of the revised manuscript as follows:

The development of flapping wing vehicles requires generic toolsets that would allow conceptual design/sizing of such vehicles while taking into consideration the correct interaction between the different subsystems in a transparent fashion. Design methods have been previously presented for different flapping wing vehicle configurations, including design tools for concepts developed for forward [42,43] and hovering [44] flight modes. Recent progress in miniaturisation of engineering systems together with the desire to create tiny aerial robots have motivated the development of a range of flapping wing vehicle concepts at insect-scale. Whilst the flapping wing motions from these concepts are generally similar, the approach through which the flapping motion is created differs between concepts, owing to the different actuators and motion amplification mechanisms adopted. In this work, we presented a framework for the conceptual design of micro DC motor-actuated insect-like flapping wing vehicles that exploit resonance as a means for successful operation.

Regarding the Reviewer's request to calculate the first type uncertainty, we have now included calculations of the uncertainty in terms of the standard error of the regression, $s_{y/x}$, for all the scaling laws developed in this work. It is to be noted that these scaling laws were originally accompanied with the calculated values of R^2 , providing a measure of the quality of the fit of the regression scaling relation. This is now complemented by the calculated standard error of regression estimate, $s_{y/x}$, which is a measure of the difference between actual values and values estimated from a given regression equation. The standard error of estimate is also defined as the standard deviation of the normal distributions of y for any given x . Hence, $s_{y/x}$ provides a measure of "class A" uncertainty. A brief description of how $s_{y/x}$ is evaluated is provided in page 4 of the revised manuscript, and all $s_{y/x}$ values are now included beside each scaling law within the revised manuscript:

$$R_w[mm] = 2.6952m_p^{0.3727}[mg], n = 12, R^2 = 0.9, s_{R_w/m_p} = 7.9mm \quad (1)$$

$$m_{w,s}[mg] = 3E^{-4}R_w^3[mm^3], n = 16, R^2 = 0.95, s_{m_{w,s}/R_w^3} = 14.3mg \quad (5)$$

$$\eta_g[\%] = 100N_g^{-0.09}, n = 7, R^2 = 0.99, s_{\eta_g/N_g} = 1.5\% \quad (19)$$

$$f_{rec}[Hz] = 19034m_a^{-0.535}[mg], n = 7, R^2 = 0.87, s_{f_{rec}/m_a} = 245Hz \quad (21)$$

$$J_m[g.cm^2] = 2 \times 10^{-8}m_a^{1.6867}[mg], n = 7, R^2 = 0.97, s_{J_m/m_a} = 0.015g.cm^2 \quad (22)$$

$$R_a[Ohm] = 1.4335m_a^{0.3578}[mg], n = 6, R^2 = 0.76, s_{R_a/m_a} = 11.90hm \quad (23)$$

$$B_m[nN.m.\frac{sec}{rad}] = 0.0084m_a^{0.8993}[mg], n = 15, R^2 = 0.87, s_{B_m/m_a} = 6.4 nN.m.\frac{sec}{rad} \quad (24)$$

$$K_a[\frac{mN.m}{A}] = 0.0012m_a^{0.9258}[mg], n = 15, R^2 = 0.89, s_{K_a/m_a} = 1.5 \frac{mN.m}{A} \quad (25)$$

$$I_0[Amp] = 0.1998m_a^{-0.359}[mg], n = 6, R^2 = 0.85, s_{I_0/m_a} = 0.0054Amp \quad (33)$$

Additionally, following the Reviewer's recommendation, A sentence is now added to the last paragraph to acknowledge error bars repeatability (a measure of variability in data) and hysteresis of used devices and sensors.

Another point to consider is that the developed design process, as any other conceptual design process, uses developed scaling laws based on currently available heuristic data. However, the level of uncertainty in technology estimates and uncertainty associated with manufacturing are currently high due to the lack of enough historical successful vehicles and actuator characteristics at this scale upon which estimation heuristics can be based. Therefore, it is necessary to note that the presented design process scaling laws should always be updated with reliable relevant data when available. **This should be done whilst considering measures for assessing the variability in data used to produce design scaling laws, together with any potential hysteresis issues of the used subsystems and sensors.**

Finally, we have added more information of the example vehicle configuration. However, we would like to stress that this paper is concerned with developing a conceptual design framework for resonant motor-driven flapping wing propulsion systems and is not specific for a certain vehicle. This is why we deliberately did not want to include lots of details on the example vehicle shown in the paper to avoid distracting readers from the main aim of the work. However, following the Reviewer's recommendation, more information on the vehicle configuration has now been added to page 3 of the revised manuscript as follows:

A video is provided in the supplementary material for the propulsion system shown in Figure 1 built using commercial off-the-shelf components. **This propulsion system weighs 3.4g. Each motor, weighing 1g, had its gear box, weighing 0.2g, fully integrated to it. An elastic element in the form of an elastic band was used. The wings had an elliptic planform shape and were realised using a polyimide film spanning on top of a rigid structure of carbon fibre rods. The passive hinge required to ensure adequate wing pitching was realised using a simple way of adding an extra carbon fibre rod parallel to the leading-edge rod. As such, the membrane material enclosed between the two rods provided the hinge functionality. This method is easy in prototyping; however, the extra carbon fibre rod adds weight to the wing structure, which is less favourable for flapping wings. Wing stoppers were added to avoid angles of attack more than 45 degrees. This propulsion system was able to demonstrate tethered operation; however, it is important to stress that this propulsion system is included here just to demonstrate the configuration and functionality of such type of concepts. In fact, this paper will focus on developing a generic conceptual design tool for resonant motor-driven flapping propulsion system and is not specific for a certain configuration.**